# Reinforcement Learning with Discrete Diffusion Policies for Combinatorial Action Spaces

Haitong Ma [* 1 2]  Ofir Nabati [* 1 3]  Aviv Rosenberg [1]  Bo Dai [4]  Oran Lang [1]  Craig Boutilier [1]  Na Li [2]
Shie Mannor [3 5]  Lior Shani [† 1]  Guy Tennenholtz [† 1]

## Abstract

Reinforcement learning (RL) struggles to scale to large, combinatorial action spaces common in many real-world problems. This paper introduces a novel framework for training discrete diffusion models as highly effective policies in these complex settings. Our key innovation is an efficient online training process that ensures stable and effective policy improvement and . By leveraging policy mirror descent (PMD) to define an ideal, regularized target policy distribution, we frame the policy update as a distributional matching problem, training the expressive diffusion model to replicate this stable target. This decoupled approa ch stabilizes learning and significantly enhances training performance. Our method achieves state-of-the-art results and superior sample efficiency across a diverse set of challenging combinatorial benchmarks, including DNA sequence generation, RL with macro-actions, and multi-agent systems. Experiments demonstrate that our diffusion policies attain comparable or superior performance compared to other baselines. Crucially, our extensive empirical analysis reveals a key trade-off: FKL demonstrates superior sample efficiency and faster initial convergence, whereas RKL ensures stable training and higher asymptotic performance on challenging tasks.

## 1. Introduction

Reinforcement learning (RL) has been instrumental in pushing the boundaries of autonomous decision-making, achieving superhuman performance in a diverse range of complex sequential tasks (Silver et al., 2016; Vinyals et al., 2019; Schrittwieser et al., 2020). However, a significant frontier remains: scaling these successes to problems with vast, combinatorial discrete action spaces. Such challenges are not niche; they are central to many real-world applications, including planning with macro-actions in hierarchical RL (Sutton et al., 1999a; Durugkar et al., 2016), coordinating strategies in multi-agent systems (Hernandez-Leal et al., 2019), and generating slates in recommender systems (Ie et al., 2019). The sheer scale of these action spaces poses a fundamental challenge to standard RL algorithms, demanding highly efficient policy parameterizations and effective exploration strategies.

Prior approaches have attempted to mitigate this complexity by mapping actions to lower-dimensional subspaces (Stulp et al., 2012; Tennenholtz & Mannor, 2019), employing hierarchical training schemes (Nachum et al., 2018), or assuming specific structural properties of the action space (Carrara et al., 2019). While effective in certain contexts, these methods often rely on structural assumptions or inductive biases that may not hold in more general and complex problem settings. More recently, the success of autoregressive models (Vaswani et al., 2017) has inspired their use for policies over combinatorial actions (Chen et al., 2021; Wen et al., 2022b). Yet, these models suffer from two key limitations: high computational cost during inference due to their sequential generation process and the imposition of a causal action ordering, which is often an artificial and restrictive constraint.

Diffusion models have emerged as a powerful class of generative models, renowned for their ability to capture highly complex probability distributions without imposing a causal structure (Sohl-Dickstein et al., 2015; Ho et al., 2020). Recent extensions to discrete spaces have further broadened their applicability (Austin et al., 2021; Sun et al., 2022; Campbell et al., 2022; Shi et al., 2024). This inherent flexibility and expressiveness make them an ideal candidate for modeling policies in large, unstructured discrete action spaces. While diffusion models have been actively explored for synthesizing policies in continuous control (Wang et al.,

---
[*]Equal contribution [†]Equal supervision [1]Google Research [2]Harvard University [3]Technion [4]Google DeepMind [5]Nvidia Research. Correspondence to: Haitong Ma and Ofir Nabati <{haitongma8,ofirnabati}@gmail.com>.

*Proceedings of the 43$^{rd}$ International Conference on Machine Learning*, Seoul, South Korea. PMLR 306, 2026. Copyright 2026 by the author(s).

2022; Ding et al., 2024; Ren et al., 2024; Ma et al., 2025), a principled and efficient framework for training discrete diffusion policies with RL remains unexplored.

In this work, we introduce a novel framework for training discrete diffusion models as highly effective policies for combinatorial action spaces. Our key innovation is an efficient online training process that ensures stable and effective policy improvement. We leverage policy mirror descent (PMD, Shani et al. (2020); Tomar et al. (2021); Lan (2023)) to define the ideal target policy distribution based on the PMD optimization objective. This reframes the policy update as a distributional matching problem, where we train our expressive diffusion model to replicate this stable target. This decoupled approach is critical: it separates the RL objective optimization from the complex task of representation learning, which we delegate to the diffusion model, thereby stabilizing the entire learning process and significantly enhancing performance.

Our core contributions are as follows: (1) We introduce RL-$D^2$, a new and efficient online training framework for using discrete diffusion models as policies in RL for combinatorial action spaces. Our core mechanism reframes the policy update as a distributional matching problem by using policy mirror descent (PMD) to define a stable target distribution, which significantly stabilizes learning. (2) We derive and analyze two practical policy improvement methods based on minimizing the forward and reverse Kullback-Leibler (KL) divergence to the PMD target. (3) Finally, we conduct extensive experiments across three distinct and challenging domains: DNA sequence generation (Gosai et al., 2023), long-horizon RL with macro-actions in Atari (Bellemare et al., 2013), and cooperative multi-agent RL in the challenging Google Research Football domain (Kurach et al., 2020). In all settings, our method achieves state-of-the-art results, demonstrating superior performance, scalability, and efficiency.

## 2. Related Work

**Discrete Diffusion Models.** Diffusion models for generating continuous data, such as images, typically rely on the gradual addition and removal of Gaussian noise to learn and synthesize complex probability distributions (Sohl-Dickstein et al., 2015; Ho et al., 2020). However, this paradigm is ill-suited for discrete data like text or biological sequences, where values are categorical and adding small amounts of continuous noise is not meaningful. To address this, *discrete diffusion models* extend the iterative refinement idea to discrete state spaces, with forward and backward processes using Markov chains where each transition in a sequence is sampled independently. Various approaches have explored different transition mechanisms and training objectives (Austin et al., 2021; Campbell et al., 2022; Sun et al., 2022; Lou et al., 2023; Shi et al., 2024). Among these, *absorbing (or masked) diffusion* has proven to be particularly effective (Sahoo et al., 2024; Ou et al., 2024). The success of these models has led to their application in a range of domains. In natural language processing, they have been adapted for complex generation tasks (Arriola et al., 2025; Ye et al., 2025; Nie et al., 2025). More relevant to our work, discrete diffusion has shown significant promise in bio-sequence modeling for generating novel proteins and DNA sequences with desired properties (Gruver et al., 2023; Wang et al., 2024a).

**Reward-based Fine-tuning and RL for Discrete Diffusion.** A key challenge, beyond unconditional generation, is adapting discrete diffusion to optimize for specific objectives. This has primarily been approached through reward-based fine-tuning, which adjusts the model's parameters to increase the likelihood of generating high-reward samples. For instance, (Wang et al., 2024a) enable direct reward backpropagation by leveraging the Gumbel-softmax trick, while (Zekri & Boullé, 2025) optimize the model by manipulating the score entropy (Lou et al., 2023).While effective, these methods can be viewed as forms of a single-step policy optimization. By contrast, the application of online RL to discrete diffusion is unexplored, and faces challenges such as exploration-exploitation trade-offs, computational efficiency, and horizon-complexity trade-off.

Black-box methods like Evolution Strategies (Salimans et al., 2017; Conti et al., 2018; Nabati et al., 2023), Genetic Algorithms (Such et al., 2017), and the Cross-Entropy Method (De Boer et al., 2005) offer alternative training paradigms, our approach provides two key advantages over standard discrete solvers. First, instead of optimizing static, single-step objectives, we leverage multi-step transition dynamics, using the discrete diffusion policy as an expressive prior to efficiently guide exploration in large combinatorial spaces. Second, our FKL variant supports off-policy training, maximizing sample efficiency through data reuse.

## 3. Preliminaries

In this section, we review the necessary background. We first define the problem setup for RL with large, combinatorial action spaces. We then introduce policy mirror descent as the foundation for our policy improvement step, followed by a review of discrete diffusion models, which will serve as our policy parameterization.

### 3.1. Problem Setup

We consider a *Markov decision process (MDP)* defined by a tuple $(\mathcal{S}, \mathcal{A}^K, P, \gamma, r, \rho_0)$, where $\mathcal{S}$ is the state space, $P$ is the transition function, $r$ is a reward function, $\gamma \in [0, 1)$ is the discount factor, and $\rho_0$ is the initial state distribution.

The action space $\mathcal{A}^K$ is assumed to be large, with some combinatorial structure, i.e., $\mathbf{a} \in \mathcal{A}^K$ is a structured, "multi-component" object [1]. The reward function $r : \mathcal{S} \times \mathcal{A}^K \to \mathbb{R}$ and transition function $P : \mathcal{S} \times \mathcal{A}^K \mapsto \Delta_{\mathcal{S}}$ are defined w.r.t. $\mathcal{A}^K$.

This general setup encapsulates a range of different problems, one of which is *hierarchical RL* (Sutton et al., 1999a; Vezhnevets et al., 2017; Haarnoja et al., 2018), where each action $\mathbf{a} \in \mathcal{A}^K$ is a *macro-action*, or a sequence of $K$ primitive actions, $\mathbf{a} = (a_1, \ldots, a_K)$.[2] In this case, $r(s, \mathbf{a})$ and $P(s'|s, \mathbf{a})$ represent the total discounted reward and the final state after executing the entire $K$-step sequence. Other examples include multi-agent policy optimization (Hernandez-Leal et al., 2019), where $\mathbf{a} = (a_1, \ldots, a_K)$ is the joint action for $K$ agents, where $a_i \in \mathcal{A}_i$ is the $i$th agent's action; slate recommendation (Ie et al., 2019), where $a_i$ is the item at the $i$th position of a set/slate of recommendations of size $K$; and combinatorial sequential assignment (Carrara et al., 2019).

A policy $\pi : \mathcal{S} \mapsto \Delta_{\mathcal{A}^K}$ maps a state to a distribution over the action space. The state-action value function $q^\pi(s, \mathbf{a})$ is the expected return after taking action $\mathbf{a}$ in state $s$ and following $\pi$ thereafter:

$$q^\pi(s, \mathbf{a}) = r(s, \mathbf{a}) + \gamma \mathbb{E}_{s' \sim P(\cdot|s, \mathbf{a}), \mathbf{a}' \sim \pi(\cdot|s')}[q^\pi(s', \mathbf{a}')]. \tag{1}$$

The state-value function is the expectation over actions, $v^\pi(s) := \mathbb{E}_{\mathbf{a} \sim \pi(\cdot|s)}[q^\pi(s, \mathbf{a})]$. The agent's goal is to find an optimal policy $\pi^*$ within a policy class $\Pi$ that maximizes the expected return: $\pi^* \in \arg\max_{\pi \in \Pi} \mathbb{E}_{s \sim \rho_0}[v^\pi(s)]$.

**Remark 1** (Compared to discrete black-box solver.)**.** *Compared with standard discrete black-box solvers, our RL-D$^2$ framework is particularly well suited to sequential decision-making problems with large combinatorial action spaces. While methods such as evolutionary algorithms and zeroth-order optimization typically treat each problem instance as a static objective and often optimize from scratch, RL-D$^2$ learns a state-conditioned diffusion policy that can exploit transition dynamics, perform structured exploration, and generalize to unseen states without re-solving each instance independently.*

### 3.2. Policy Mirror Descent

*Policy mirror descent (PMD)* (Beck & Teboulle, 2003; Shani et al., 2020; Tomar et al., 2020; Lan, 2023) is a policy

---

[1]For simplicity we focus on power sets of $\mathcal{A}$, though more complex combinatorial action spaces can be used.

[2]We abuse terminology slightly. In general, macro-actions (or *options*) are general "local" policies with suitable termination conditions that can used within a larger hierarchical or abstract policy (Sutton et al., 1999b; Hauskrecht et al., 1998). However, the fixed sequence view of macros (a special case of the former) also appears in the literature (Durugkar et al., 2016).

optimization method that provides a provably convergent stable and regularized policy improvement step. Given a current policy $\pi_{\text{old}}$, the PMD update finds a new policy $\pi$ by solving:

$$\pi(\cdot|s) \in \arg\max_{\pi \in \Pi} \mathbb{E}_{\mathbf{a} \sim \pi(\cdot|s)}[A^{\pi_{\text{old}}}(s, \mathbf{a})] - \lambda d_{KL}(\pi, \pi_{\text{old}}; s) \tag{2}$$

$\forall s \in \mathcal{S}$, where $A^{\pi_{\text{old}}}(s, \mathbf{a}) := q^{\pi_{\text{old}}}(s, \mathbf{a}) - v^{\pi_{\text{old}}}(s)$ is the advantage function, $\lambda > 0$ is a temperature parameter, and $d_{KL}$ is the Kullback-Leibler (KL) divergence: $d_{KL}(\pi, \mu; s) := \sum_{\mathbf{a} \in \mathcal{A}^K} \pi(\mathbf{a}|s) \log \frac{\pi(\mathbf{a}|s)}{\mu(\mathbf{a}|s)}$.

The unique solution to this optimization problem is given by:

$$\pi_{\text{MD}}(\mathbf{a}|s) = \pi_{\text{old}}(\mathbf{a}|s) \exp(A^{\pi_{\text{old}}}(s, \mathbf{a})/\lambda) \, / \, Z(s), \tag{3}$$

where $Z(s) = \mathbb{E}_{\mathbf{a} \sim \pi_{\text{old}}}[\exp(A^{\pi_{\text{old}}}(s, \mathbf{a})/\lambda)]$ is the normalization constant, or partition function.

### 3.3. Masked Discrete Diffusion Processes

We provide a general background on discrete diffusion in this subsection, and refer the reader to Austin et al. (2021) and Appendix A for an exhaustive derivation of this method. A reader already familiar with discrete diffusion processes can skip directly to Sec. 4.

Discrete diffusion models are powerful generative models, well-suited for capturing complex distributions over structured, sequential data. We therefore focus on combinatorial action spaces that can be represented as a fixed-length sequence of $K$ discrete actions, $\mathbf{a} = (a_0, \ldots, a_{K-1}) \in \mathcal{A}^K$. This formulation directly applies to the macro-action problem and can be adapted for other settings like multi-agent joint actions by imposing a consistent ordering on the agents. We use a masked diffusion process (Shi et al., 2024), which operates over an augmented vocabulary $\mathcal{A} \cup \{\mathtt{m}\}$ that includes a mask action $m$.

**Forward Process.** The fixed forward process $q$ gradually noises a clean macro-action $\mathbf{a}^0 \in \mathcal{A}^K$ into a fully masked sequence $\mathbf{a}^N$ over $N$ discrete steps. This noising process is applied independently to each component $a_t \in \mathbf{a}$. The single-action transition is defined as

$$q\left(a^n \mid a^{n-1}\right) = \begin{cases} \beta_n & \text{if } a^{n-1} \neq \mathtt{m} \text{ and } a^n = \mathtt{m} \\ 1 - \beta_n & \text{if } a^{n-1} \neq \mathtt{m} \text{ and } a^n = a^{n-1} \\ 1 & \text{if } a^{n-1} = a^n = \mathtt{m} \\ 0 & \text{otherwise} \end{cases},$$

where $\{\beta_n\}_{n=1}^N$ is a fixed schedule. This defines a marginal distribution

$$q(a^n|a^0) = \text{Cat}\left(a^n; \alpha_n a^0 + (1 - \alpha_n)\mathbf{e}_m\right),$$

where $a^n = a^0$ with probability $\alpha_n$ and $a^n = m$ with probability $1 - \alpha_n$, for a known noise schedule $\alpha_n$.

**Reverse Process.** The learned reverse process $p_\theta(\mathbf{a}^{n-1}|\mathbf{a}^n, s)$ is trained to reverse the forward process, conditioned on the state $s$. It iteratively denoises a sequence $\mathbf{a}^n$, starting from the pure noise prior $\mathbf{a}^N \sim p(\cdot|s)$, to generate a clean macro-action $\mathbf{a}^0 \sim \pi_\theta(\cdot|s)$. This process is parameterized by the posterior:

$$
\begin{aligned}
&q(a^{n-1}|a^n, a^0) \\
&= \begin{cases} \mathrm{Cat}\left(a^{n-1}; \bar{\alpha}_n a^0 + (1-\bar{\alpha}_n)\mathbf{e}_m\right) & a^n = \mathbf{e}_m \\ \mathrm{Cat}\left(a^{n-1}; a^n\right) & a^n \neq \mathbf{e}_m \end{cases}
\end{aligned}
$$

where $\bar{\alpha}_n := \frac{\alpha_{n-1} - \alpha_n}{1 - \alpha_n}$, through the parameterized model $p_\theta(a^{n-1}|a^n) := q(a^{n-1}|a^n, \mu_\theta(a^n, n))$, where

$$
\mu_\theta(a^n, n) = \begin{cases} \mathrm{softmax}(f_\theta(a^n, n)) & a^n = \mathtt{m}, \\ a^n & a^n \neq \mathtt{m}. \end{cases}
$$

is the clean sample mean-value estimator induced by a trained model $f_\theta$ (e.g., a Transformer) that predicts the clean sequence $\mu_\theta(\mathbf{a}^n, n, s) \approx \mathbf{a}^0$ from any noised sequence $\mathbf{a}^n$ at step $n$.

**Training Objective.** The model $f_\theta$ is trained by maximizing the Evidence Lower Bound (ELBO), $\mathcal{L}_{\mathrm{ELBO}}(\mathbf{a}^0, s; \theta)$, which is a lower bound on the log-likelihood $\log \pi_\theta(\mathbf{a}^0|s)$. This objective trains the network to reconstruct the clean action $\mathbf{a}^0$ from its noised versions $\mathbf{a}^n$. For a macro-action $\mathbf{a}^0$ with $K$ actions, the objective to maximize is a sum of weighted negative cross-entropy terms over all diffusion steps $n$ and actions $k$:

$$
\begin{aligned}
&\mathcal{L}_{\mathrm{ELBO}}(\mathbf{a}^0, s; \theta) = \\
&\sum_{n=1}^N \bar{\alpha}_n \mathbb{E}_{\mathbf{a}^n \sim q(\cdot|\mathbf{a}^0)} \Big[ \sum_{k=0}^{K-1} \delta_{a_k^n, m} \cdot \log \mu_\theta(\mathbf{a}^n, n, s)_{a_k^0} \Big]
\end{aligned} \tag{4}
$$

where $\bar{\alpha}_n$ is a weighting term derived from the noise schedule, $\delta_{a_k^n, m}$ is an indicator function that is 1 if the $k$-th action is masked (and 0 otherwise), and $\log \mu_\theta(\cdot)_{a_k^0}$ is the model's predicted log-probability for the original clean action $a_k^0$. The full derivation is detailed in Appendix A.

# 4. RL-D$^2$: Reinforcement Learning with Discrete Diffusion

We now introduce our framework for training discrete diffusion policies. Our approach follows a policy iteration structure. Let $k$ be the current training iteration. The process alternates between (1) policy evaluation, which estimates the Q-function $q^{\pi_k}$ for the current policy $\pi_k$, and (2) policy improvement. The core of our method lies in the latter improvement step.

We first define a target distribution, $\pi_{\mathrm{MD}}^k$, which is the mirror descent iteration optimal solution from Eq. (3) calculated

using $\pi_{\mathrm{old}} \equiv \pi_k$ and $q^{\pi_{\mathrm{old}}} \equiv q^{\pi_k}$. This transforms the policy improvement problem into a distributional approximation problem, a common paradigm in deep RL (Chan et al., 2022; Abdolmaleki et al., 2018). The new policy $\pi_{k+1}$ is then obtained by finding the parameters $\theta$ that minimize a chosen divergence $d$ to this target; namely,

$$
\pi_{k+1} \in \arg\min_{\pi_\theta \in \Pi} \mathbb{E}_{s \sim \mathcal{D}} \big[ d(\pi_\theta, \pi_{\mathrm{MD}}^k; s) \big] \quad \text{(IMPR. STEP)}
$$

where $\mathcal{D}$ is a distribution of states, typically from a replay buffer or current policy stationary state distribution. A flowchart summarizing our approach is presented in Fig. 1.

## 4.1. Reverse and Forward KL

The choice of the divergence $d$ in (IMPR. STEP) is critical and defines the practical update rule. We focus on the Kullback-Leibler (KL) divergence (i.e., $d \equiv d_{KL}$). Specifically, we consider two variants of (IMPR. STEP) using forward KL and reverse KL divergences, which result in two different methods for policy improvement, as we explain below. We refer the reader to (Chan et al., 2022) for a thorough review of reverse and forward KL properties in RL.

**Forward KL Divergence (FKL).** The forward KL objective, $d_{KL}(\pi_{\mathrm{MD}}^k, \pi_\theta; s)$, seeks a policy $\pi_\theta$ that covers the modes of the target distribution. This "mean-seeking" behavior can be beneficial for exploration, as it encourages the policy to maintain probability mass over all high-value actions (Chan et al., 2022). Minimizing this objective directly is intractable. Instead, we minimize a tractable bound derived by applying the diffusion model's ELBO inequality to the KL definition (see Appendix B.1). This results in the following weighted ELBO loss:

$$
\begin{aligned}
&\mathcal{L}_{\mathrm{FKL}}(\theta) = \\
&- \mathop{\mathbb{E}}_{\substack{s \sim \mathcal{D} \\ \hat{\mathcal{A}}_s \sim \pi_k}} \Big[ \sum_{\substack{\mathbf{a}^0 \in \hat{\mathcal{A}}_s}} (\mathrm{softmax}_{\mathbf{a} \in \hat{\mathcal{A}}_s}(A^{\pi_k}(s, \mathbf{a}^0)/\lambda)) \cdot \mathcal{L}_{\mathrm{ELBO}}(\mathbf{a}^0, s; \theta) \Big].
\end{aligned}
$$
$$\text{(FKL Loss)}$$

Here, $\hat{\mathcal{A}}_s$ is a batch of macro-actions sampled from the "old" policy $\pi_k$ (i.e., a target network, $\pi_{\theta_{\mathrm{old}}}$), and $\mathcal{L}_{\mathrm{ELBO}}$ is the weighted cross-entropy loss in equation 4. The softmax re-weights the sampled actions to approximate the target distribution $\pi_{\mathrm{MD}}^k$. This objective effectively trains the diffusion model as a generative classifier, focusing the model's capacity on reconstructing high-value actions more frequently and is more easily adjusted to off-policy training.

**Reverse KL Divergence (RKL).** The reverse KL objective, $d_{KL}(\pi_\theta, \pi_{\mathrm{MD}}^k; s)$, is equivalent to the original PMD optimization in Eq. (2) (see Appendix B.2). This objective has theoretical policy improvement guarantees (Chan et al.,

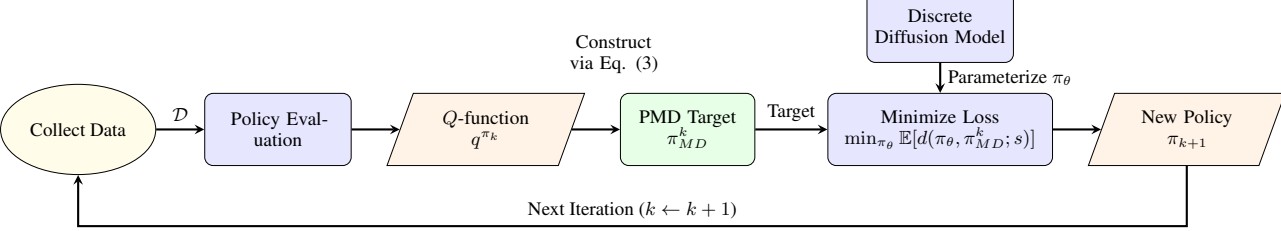

*Figure 1.* **Overview of the RL-D$^2$ Framework.** Our framework adopts a policy iteration structure. Following policy evaluation, which estimates the current Q-function, the policy improvement step is implemented as a distributional matching problem. Here, the discrete diffusion policy is trained to minimize the KL divergence (FKL or RKL) relative to an optimal target distribution ($\pi_{MD}^k$) derived via Policy Mirror Descent in equation 3.

2022) and results in a "mode-seeking" policy that focuses on the highest-value action. This objective can be written as follows:

$$\mathcal{L}_{\mathrm{RKL}}(\theta) =$$
$$\mathbb{E}_{s\sim\mathcal{D},\mathbf{a}\sim\pi_k}[-\eta(s,\mathbf{a};\theta)A^{\pi_k}(s,\mathbf{a}) + \lambda d_{KL}(\pi_\theta,\pi_k;s)],$$
$$\text{(RKL Loss)}$$

where here, $\eta(s,\mathbf{a};\theta) := \pi_\theta(\mathbf{a}|s)/\pi_k(\mathbf{a}|s)$ is an importance sampling (IS) ratio. In this case, $\mathcal{D}$ usual choice is the state occupancy measure of $\pi_k$ (Schulman, 2015; Shani et al., 2020) for an on-policy training.

For diffusion policies, the likelihood $\pi_\theta(\mathbf{a}|s)$ is intractable, and thus so is the ratio $\eta$. Following Ren et al. (2024), we can construct an augmented MDP where states are $(s,\mathbf{a}^n)$ (an environment state and a noisy action at diffusion step $n$) and the advantage for a denoising step is defined by the final clean action's advantage, i.e., $A^{\pi_k}((s,\mathbf{a}^n),\mathbf{a}^{n-1}) \triangleq A^{\pi_k}(s,\mathbf{a}^0)$. This yields a tractable IS ratio based on the single-step reverse process:

$$\eta((s,\mathbf{a}^n),\mathbf{a}^{n-1};\theta) = \frac{p_\theta(\mathbf{a}^{n-1}|\mathbf{a}^n,s)}{p_k(\mathbf{a}^{n-1}|\mathbf{a}^n,s)}.$$

We refer to this ratio as "single-step ratio". Given a tractable estimator for $\eta$, we optimize the objective in Eq. (RKL Loss) using a PPO-style clipping mechanism (Schulman et al., 2017).

### 4.2. On-Policy Diffusion Learning

The standard ELBO objective, used in both FKL Loss and RKL Loss, trains the model to denoise samples $(\mathbf{a}^n,\mathbf{a}^0)$ generated from the *fixed forward process* $q(\mathbf{a}^n|\mathbf{a}^0)$. However, this distribution of noised actions $\mathbf{a}^n$ may differ significantly from the actions the policy actually generates during its own generative process. To align the training distribution with the inference distribution, we propose *On-Policy Diffusion Learning*. Instead of starting from a clean action $\mathbf{a}^0 \sim \pi_k$ and adding noise, we generate the entire diffusion trajectory $(\mathbf{a}^N,\ldots,\mathbf{a}^0)$ on-policy by sampling from the *learned reverse process* of the current policy, i.e., $\mathbf{a}^N \sim p(\cdot|s)$ and

$\mathbf{a}^{n-1} \sim p_{\theta_k}(\cdot|\mathbf{a}^n,s)$. This yields $(\mathbf{a}^n,\mathbf{a}^0)$ pairs that are "on-policy" with respect to the policy's own generative dynamics, which we find enhances stability and sample efficiency. Note that the "on-policy" here only refers to the diffusion process, rather than the overall RL framework.

## 5. Experiments

We conduct a comprehensive set of experiments to evaluate our proposed framework for training discrete diffusion policies. Our evaluation spans three distinct and challenging domains to demonstrate the method's effectiveness, scalability, and versatility:

1. Reward-based finetuning for DNA sequence generation.

2. Online reinforcement learning with long-horizon in complex single-agent Atari environments.

3. Multi-agent cooperative learning with combinatorial joint action spaces.

### 5.1. DNA Sequence Generation: Single-Step Policy Optimization

We first validate our approach on a reward-guided generation task, which serves as a single-step RL problem (i.e., combinatorial multi-armed bandit). The goal is to finetune a pretrained discrete diffusion model to generate DNA sequences that maximize a specific reward signal, verifying the effectiveness of our policy optimization algorithm.

We use a large public enhancer dataset of approximately 700,000 DNA sequences with length 200 (Gosai et al., 2023). A reward function, detailed in Appendix D.1, is defined to predict gene expression activity, we leverage the pre-trained reward model provided by (Wang et al., 2024a). Our primary metrics are the reward achieved and the approximate log-likelihood of the generated sequences, which measures their naturalness. We compare against controlled generation methods such as conditional guidance (CG) (Nisonoff et al., 2024), SMC and TDS (Wu et al., 2023) and classifier-free

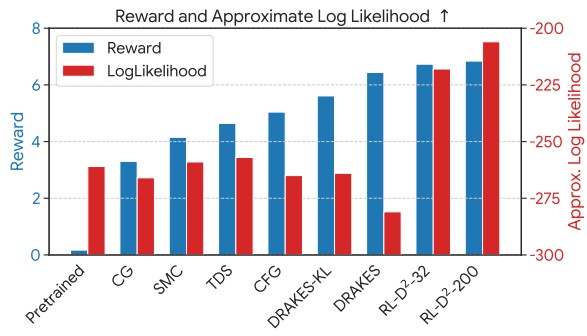

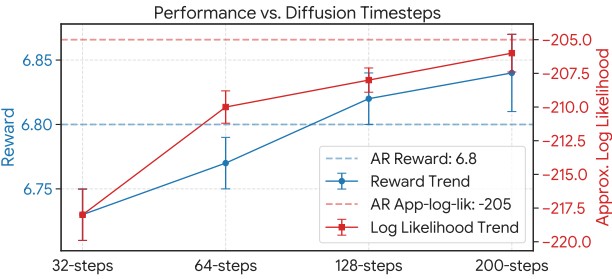

*Figure 2.* Reward and Approximate Likelihood of DNA generation. **Left:** The proposed RL-D$^2$ gets best performance on reward and log likelihood, even with fewer diffusion time steps. **Right:** The mean and 95% confidence intervals of reward and approximate log likelihood with various diffusion timesteps.

guidance (CFG) (Ho & Salimans, 2022), as well as a strong RL-based baseline, DRAKES (Wang et al., 2024a), that optimizes the policies by backpropagating reward through the reverse process using Gumbel-Softmax trick. For this task, we optimize our policy using the forward KL (FKL) objective Eq. (FKL Loss).

As shown in Fig. 2 (left), our method achieves a new state-of-the-art, attaining the highest reward scores while simultaneously generating the most probable sequences (highest log-likelihood). This demonstrates that our FKL-based update effectively optimizes for the target reward without sacrificing generative quality. Moreover, Fig. 2 (right) showed that we can achieve consistently strong performance with diffusion timesteps much smaller than the sequence length 200, highlighting the inference-time efficiency compared to autoregressive (AR) generation. Finally, our approach is significantly more computationally efficient than DRAKES. As we don't need to backpropagate through the whole reverse process, we reduced GPU memory consumption from 66.4 GB to 10.6 GB, and computation time from 268 minutes to 97 minutes compared to DRAKES, making high-performance reward optimization more accessible.

## 5.2. Reinforcement Learning with Macro Actions

Next, we evaluate our RL-D$^2$ in the challenging MinAtar (Young & Tian, 2019) and Atari (Bellemare et al., 2013) benchmarks, where the agent learns to make deci-

sions over long horizons by generating macro-actions, i.e., sequences of primitive actions. Our experiments are designed to assess the performance and scalability in using diffusion policies for complex planning tasks. For these tasks, we optimize our policy using the forward KL (FKL) objective in Eq. (FKL Loss).

We evaluate RL-D$^2$ on the MinAtar benchmark (Young & Tian, 2019), a suite of simplified Atari games that provide a controlled setting without partial observability. We employ a **macro-action length of** 4. We compare against DQN (Mnih et al., 2015), IMPALA (Espeholt et al., 2018), and their macro-action-enabled variants (*DQN-Macro*, *IMPALA-Macro*). In *DQN-Macro*, the $Q$-network's output dimension is modified to $|\mathcal{A}|^4$ to select one of all possible length-4 macro-actions. For *IMPALA-Macro*, the policy network's output is changed to $4 \times |\mathcal{A}|$, allowing it to sample the four actions of the macro-action independently at once. The detailed implementations are in Appendix D.2.

As shown in Table 1, RL-D$^2$ achieves substantially stronger performance in 2 out of 5 tasks in MinAtar. The substantial score improvements in BREAKOUT and SPACE INVADERS mean that the agents can keep alive for a very long time, highlighting the policy's ability to discover and represent complex, long-term strategies. While standard baselines adapted for macro-actions show modest gains, they are far outstripped by our approach, underscoring the necessity of an expressive generative model to effectively navigate large combinatorial action spaces.

We confirm our findings on the full Atari benchmark (Bellemare et al., 2013) with additional strong baselines including R2D2 (Kapturowski et al., 2018) and PPO (Schulman et al., 2017). As shown in Figure 3, our method achieves the highest average human-normalized score in average and outperforms strong baselines using both macro and single actions in 36 of 56 environments (the full results can be found in Appendix E.2), showcasing the strong performance of the proposed discrete diffusion. We start with comparing FKL against RKL in order to examine which performs better, followed by testing the scalability and horizon-complexity trade-off of our diffusion policy. Lastly, we demonstrate our diffusion policies performance with different sampling techniques.

**Comparing FKL vs. RKL with Same Computation.** The actual bottleneck of Atari games is computations due to using a deep residual network to handle image inputs. Therefore, to compare the performance and stability of FKL and RKL in Atari games, we align the training wall time while adapting the batch size. The results are shown in Fig. 4. We can see that in general FKL has a better performance than RKL. This observation is re-verified later in Google Research Football environment (Sec. 5.3). Moreover, FKL is less sensitive batch sizes, showing only marginal in-

*Table 1.* MinAtar performance. Mean and 95% bootstrap confidence intervals of scores over the last 100 evaluation episodes with 20 random seeds during training on MinAtar.

| | ASTERIX | BREAKOUT | FREEWAY | SEAQUEST | SPACE INVADERS |
|---|---|---|---|---|---|
| DQN | **284.38** [112.88, 455.88] | 180.36 [64.23, 296.49] | **60.44** [58.46, 62.42] | 109.20 [81.23, 137.17] | 1.16K [0.51K, 1.81K] |
| DQN-MACRO | 43.05 [32.68, 53.42] | 315.85 [111.90, 519.80] | 58.42 [53.51, 63.33] | **166.65** [109.25, 224.05] | 769.86 [258.27, 1281.45] |
| IMPALA | 24.78 [17.38, 32.18] | 1.02 [1.02, 1.02] | 41.01 [26.90, 55.12] | 44.55 [30.95, 58.15] | 47.71 [47.71, 47.71] |
| IMPALA-MACRO | 21.29 [12.68, 29.90] | 7.21 [6.78, 7.64] | 54.09 [44.37, 63.81] | 55.42 [32.49, 78.35] | 33.18 [33.18, 33.18] |
| RL-D$^2$ (OURS) | 52.38 [44.59, 60.17] | **20.58K** [13.47K, 27.46K] | 59.98 [57.70, 62.26] | 164.20 [109.21, 219.19] | **184.3K** [62.56K, 305.9K] |

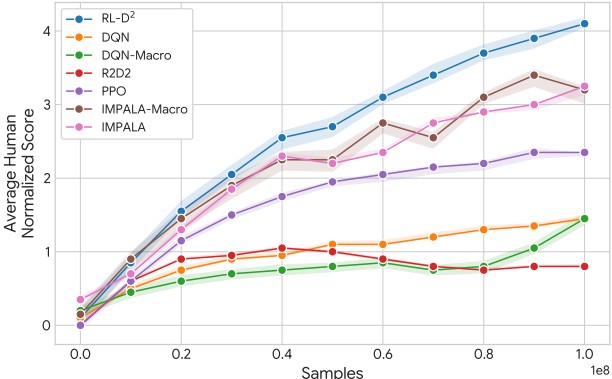

*Figure 3.* Atari performance. Performance improvement over the best baseline, evaluated by the percentage of human normalized scores.

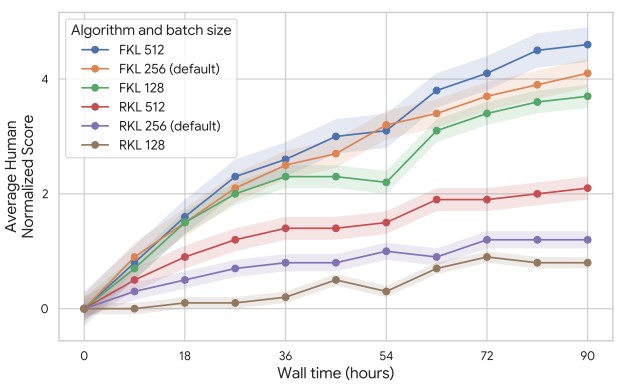

*Figure 4.* Comparing FKL and RKL, on-policy, in Atari games with different learner batch sizes. The curves shows averaged human normalized score over 10 Atari games with 3 random seeds each game.

crease when increasing batch sizes. RKL show large performance improvement when increasing batch size from 256 to 512[3], showing RKL favors larger batches because the high-variance nature of augmented MDP method.

**Scalability and horizon-complexity trade-off.** We investigate the scalability to solve more challenging MDPs by varying the macro-action length. The difficulties of sloving the MDPs increase because the size of the action space grows exponentially with respect to the macro action length, while the horizon only shrinks linearly. Fig. 5 shows that with a fixed computational budget, performance peaks at a macro-action length of 4. However, the key advantage of our method is its scalability. As shown in Fig. 5, when we scale up model capacity and data proportionally to the action space complexity (details setup listed in Appendix D.3), our diffusion policy's performance continues to improve, and performs at least as well as IMPALA-Macro up to macro action length 8, and outperforms IMPALA-Macro at macro action length 16. *DQN-Macro* fails to fit in a reasonable learner with macro action 8 and 16 as the size of the action space increases exponentially, while the baseline *IMPALA-Macro* fails to achieve better performance when the macro

action length increases from 8 to 16. Detailed performance of RL-D$^2$ with increasing computational resources from macro action length 8 to 16 are listed in Fig. 5.

**Efficient and flexible sampling techniques.** We evaluate the inference-time efficiency and flexibility of discrete diffusion policies. To make the difference clear, we extend to a long macro action setup with length 32. We leverage two techniques to further improve the sampling qualities of discrete diffusion models for these extra-long macro actions, **(1)** Top-p sampling or nuclear sampling (Holtzman et al., 2019) that selects actions from the smallest set of actions whose cumulative probabilities exceed a certain threshold; **(2)** Remasking diffusion process (Wang et al., 2025) that allows the actions to be re-masked and re-unmasked during the reverse process. The implementation details can be found in Appendix C.3.

As seen in Figure 6, top-p sampling enhanced the performance of RL-D$^2$ with fewer diffusion steps, such as 4 and 8, making inference-time more efficient without losing performance. Remasking sampling performs best when the number of timesteps is close to the sequence length. This highlights the flexibility of diffusion models.

**More Experimental Results and Ablation studies.**

In the appendix, we explore other techniques and perform

---

[3]One sample in the batch is a 32-step trajectory rather than a transition pair, which is a common practice in training pipelines used for Atari games. Therefore, the actual number of transition pairs is 32 times more.

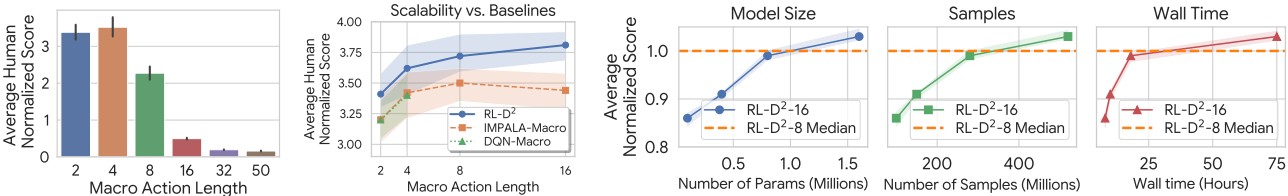

*Figure 5.* **Atari Average Human Normalized Score. Left:** Mean and 95% confidence intervals of averaged episode return over all 56 tasks to show the trade-off between planning horizon and model complexity with fixed network size and data. **Middle:** The proposed method scales more effectively with increasing network size and data compared to baselines. *DQN-Macro* fails to learn in a reasonable amount of time as the action space grows too large with macro actions more than 4. **Right:** Mean episode return of RL-D$^2$ with 16 macro actions compared to the 8 macro actions as a function of model parameters, data samples, and training time, averaged over 4 tasks and 3 seed each.

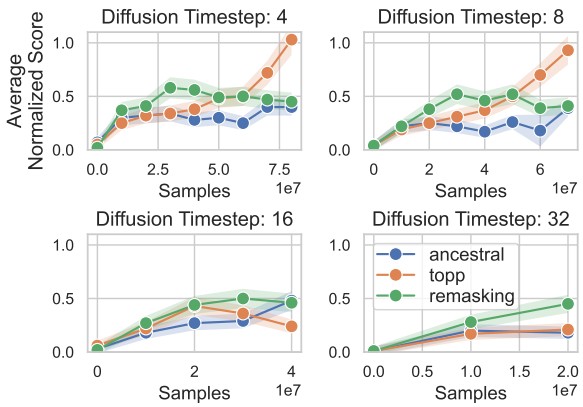

*Figure 6.* Mean and 95% confidence intervals of scores averaged over 4 tasks and 3 seed each normalized by best scores achieved with macro action length 32 as a function of diffusion timesteps and sampling techniques. Top-p sampling excels with very few steps while remasking improve the performance with more diffusion steps.

ablation studies. This includes **(1)** automatically tuning the temperature parameter $\lambda$ in Eq. (3) by enforcing a hard KL constraint; **(2)** leveraging discrete diffusion as a planner instead of committing to all macro actions generated, and **(3)** an ablation study of on-policy diffusion training discussed in Sec. 4.2. Please refer to Appendix E for the results and discussions.

### 5.3. Cooperative Multi-Agent Reinforcement Learning

Finally, we evaluate our framework in cooperative multi-agent reinforcement learning (MARL), where the combinatorial action space is the joint action of all agents. Efficiently searching this space is a primary challenge in MARL. By modeling the joint action distribution, our diffusion policy can capture complex inter-agent dependencies without relying on restrictive factorization assumptions.

We test our policy on the challenging Google Research Football benchmark (Kurach et al., 2020), using scenarios ranging from small-scale tasks (*3 vs 1*) to full-team games

(*11 vs 11*). Our diffusion model generates the joint action for all controlled agents simultaneously. We adopt the feature extractor and scenario settings suggested by (Song et al., 2023); for further implementation details, please refer to appendix D.4.

We evaluate our diffusion policy, examining both RKL and FKL objective variants against an autoregressive sampler baseline. Additionally, we compare our method against two centralized MARL policy baselines: Multi-Agent PPO (MAPPO) (Yu et al., 2022) and the **current state-of-the-art**, Multi-Agent Transformer (MAT) (Wen et al., 2022b). Note that MAT operates autoregressively, mitigating causal bias by training over random permutations of the agents' order. Finally, to assess sample efficiency, we conduct an ablation on the full game scenario (11 vs. 11) using significantly smaller budget of 100M environment steps (compared to the standard 1G) - marked as *11 vs 11 sm*.

As presented in Table 2, our discrete diffusion policy variants achieve the highest mean win rates across all scenarios. This advantage is particularly presented in the most challenging tasks requiring intricate team coordination, such as *5 vs 5*, *corner*, *ct-hard*, and the highly complex *11 vs 11* full game, where our method shows a clear advantage over state-of-the-art methods. regarding the baselines, we note that the naive autoregressive sampler lags significantly behind.

This suggests that to match the performance of diffusion models, autoregressive methods require additional alignment techniques, such as the random permutations used in MAT, to mitigate causal bias. Furthermore, while the RKL objective yields superior results overall, FKL significantly outperforms RKL (and other baselines) in the small-budget regime *11 vs 11 sm*, approaching a 70% win rate. RKL is more exploratory, allowing it to achieve optimal behavior given sufficient samples, whereas the FKL exploits faster at the cost of less exploration. Moreover, we observe FKL is likely to collapse if the temperature tuning is not handled well (results shown in Appendix E.3.3). Overall, these

*Table 2.* Final performances on Google Research Football academy scenarios, measured by maximum mean win rate scaled by 100 with the bootstrap confidence intervals over 20 random seeds. Methods with the highest mean and those that have overlapped confidence intervals with the highest mean are **bolded**,

| Method | Pass & Shoot | Run Pass & Shoot | 3 vs 1 | CT-Easy | CT-Hard | Corner | 5 vs 5 | 11 vs 11 | 11 vs 11 SM |
|---|---|---|---|---|---|---|---|---|---|
| Autoregressive | 79.4 [44.2, 94.5] | 68.9 [36.4, 89.3] | 74.1 [43.4, 91.7] | 70.8 [43.0, 89.5] | 35.2 [0.5, 66.5] | 0.0 [0.0, 0.0] | 72.5 [46.0, 90.1] | 33.1 [19.8, 46.4] | 0.0 [0.0, 0.0] |
| RL-D$^2$ RKL (ours) | **100.0 [100.0, 100.0]** | **97.9 [96.3, 99.1]** | **99.2 [98.5, 99.7]** | **98.5 [97.7, 99.2]** | **98.2 [96.3, 99.4]** | **97.1 [96.3, 98.9]** | **99.1 [98.7, 99.5]** | **99.4 [99.0, 99.7]** | 0.0 [0.0, 0.0] |
| RL-D$^2$ FKL (ours) | **98.4 [97.3, 99.3]** | **99.2 [98.7, 99.6]** | 96.8 [94.8, 98.5] | **97.4 [96.2, 98.6]** | **96.5 [93.7, 98.6]** | 94.1 [91.7, 96.1] | 97.9 [96.8, 98.5] | 98.1 [97.1, 99.0] | **68.5 [62.0, 75.0]** |
| MAT | **96.8 [94.6, 98.6]** | **97.6 [96.3, 98.8]** | 91.8 [90.6, 93.0] | 88.5 [86.4, 90.6] | 87.1 [83.5, 90.7] | 94.6 [92.1, 95.9] | 92.9 [91.9, 93.9] | 91.3 [88.9, 93.7] | 10.2 [7.4, 13.0] |
| MAPPO | 98.8 [98.4, 99.2] | 74.5 [71.6, 77.4] | 92.6 [91.0, 94.2] | 71.4 [68.3, 74.5] | 64.2 [62.1, 66.3] | 54.7 [50.2, 59.2] | 94.8 [93.1, 96.5] | 53.9 [50.9, 56.9] | 4.8 [4.5, 5.1] |

results highlight the potential of discrete diffusion models to effectively generate highly coordinated joint actions in challenging multi-agent tasks.

### 5.4. Empirical Selection of FKL and RKL

We summarize our empirical observations about FKL or RKL to provide guidance to practitioners from the viewpoint of balancing the trade-off between computational efficiency and asymptotic performance.

**Data efficiency**: FKL demonstrates faster learning in the initial stages with fewer samples, as evidenced by the *11 vs 11 sm* scenario in Table 2 and the Atari benchmarks in Fig. 4. We attribute this to the temperature values employed. As shown in Fig. 11, FKL performs well only with a large KL constraint and small temperatures; this configuration promotes aggressive exploitation by rapidly shifting the policy toward being greedy and deterministic. In contrast, our RKL implementation utilizes importance sampling ratio clipping. This mechanism results in more gradual shifts in the policy distribution, thereby leading to a slower learning pace.

**Asymptotic performance and stability.** Table 2 demonstrates that RKL gets comparable or superior performance compared to FKL when trained with large sample sizes. Moreover, the temperature schedule ablation for Google Football in Fig. 13 indicates that FKL is prone to collapse if the temperature is not appropriately selected. This slightly inferior performance of FKL may stem from two factors: (1) FKL optimizes the ELBO, a lower bound of policy loss, whereas RKL optimizes the unbiased policy mirror descent loss; and (2) the potential for collapse is caused by the fact that policy being too deterministic results in a lack of diversity in the replay buffer.

In summary, our empirical results have shown that, RL-D$^2$-FKL favors learning faster by heavy exploitation, but is less stable and requires careful temperature tuning. It is suitable for tasks with data and computational bottleneck such as Atari games. RL-D$^2$-RKL learns slightly slower but is more stable and have better asymptotic performance. It is suitable for tasks with cheap sampling cost, such as Google Football.

## 6. Summary

This paper introduces RL-D$^2$, a novel framework for reinforcement learning with discrete diffusion policies, aimed at solving decision making problems with large, combinatorial action spaces. In this framework, we propose to train diffusion models by fitting their output distribution to the analytic solution of the Policy Mirror Descent policy optimization algorithm. This is done by projecting the outputs of the model with either the forward or reverse KL divergences. Extensive experiments demonstrate that the proposed method achieves state-of-the-art performance across three challenging domains: reward-guided sequence generation, long-horizon planning with macro-actions, and cooperative multi-agent RL. The results suggest that the RL-D$^2$ framework provides a scalable and high-performing solution to a long-standing challenge in RL, effectively handling complex, combinatorial action spaces where traditional methods often fail.

## Impact Statement

This paper presents work whose goal is to advance the field of machine learning. There are many potential societal consequences of our work, none of which we feel must be specifically highlighted here.

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

## A. Masked Discrete Diffusion Process

Similar to the continuous diffusion, discrete diffusion is also composed by a fixed forward process and a learned reverse process. The forward process degrades a data sample $x^0 \sim q\left(x^0\right)$ into a sequence of progressively noisier latent variables $x^1, x^2, \ldots, x^N$ via a Markov chain,

$$q\left(x^{1:N} \mid x^0\right) = \prod_{n=1}^{N} q\left(x^n \mid x^{n-1}\right), \text{where} \quad q\left(x^n \mid x^{n-1}\right) = \text{Cat}\left(x^n; p = \mathbf{Q}_n^\top x^{n-1}\right)$$

where $\mathbf{Q}_n$ is the transition matrix with $[\mathbf{Q}_n]_{ij} = q\left(x^n = j \mid x^{n-1} = i\right)$. Specifically, we focus on a family of discrete diffusion processes called masked diffusion models (Austin et al., 2021; Campbell et al., 2022; Shi et al., 2024), where an additional `[MASK]` action is added to the action space. Denote the mask action as a special m action, the transition kernel of the forward process is defined as

$$\mathbf{Q}_n = (1 - \beta_n)\mathbf{I} + \beta_n \mathbf{1}\mathbf{e}_m^\top,$$

In another word,

$$q\left(x^n \mid x^{n-1}\right) = \begin{cases} \beta_n & \text{if } x^{n-1} \neq \text{m and } x^n = \text{m} \\ 1 - \beta_n & \text{if } x^{n-1} \neq \text{m and } x^n = x^{n-1} \\ 1 & \text{if } x^{n-1} = x^n = \text{m} \\ 0 & \text{otherwise} \end{cases}$$

which means the action is masked out with $1 - \beta_n$ probability, otherwise stays as the same. The masking schedule is defined as $\alpha_n := \prod_{i=1}^{n}(1 - \beta_i)$. Once the action is masked, it stays as the masked action m. The learned reverse Markov process $p_\theta(x^{0:N}) = p(x^N) \prod_{n=1}^{N} p_\theta(x^{n-1}|x^n)$ gradually denoises (unmasks) the latent variables towards the data distribution. The reversal model estimates the posterior:

$$q(x^{n-1}|x^n, x^0) = \begin{cases} \text{Cat}\left(x^{n-1}; \bar{\alpha}_n x^0 + (1 - \bar{\alpha}_n)\mathbf{e}_m\right) & x^n = \mathbf{e}_m, \\ \text{Cat}\left(x^{n-1}; x^n\right) & x^n \neq \mathbf{e}_m \end{cases}$$

where $\bar{\alpha}_n := \frac{\alpha_{n-1} - \alpha_n}{1 - \alpha_n}$, through the parameterized model $p_\theta(x^{n-1}|x^n) := q(x^{n-1}|x^n, \mu_\theta(x^n, n))$, where

$$\mu_\theta(x^n, n) = \begin{cases} \text{softmax}(f_\theta(x^n, n)) & x^n = \text{m}, \\ x^n & x^n \neq \text{m}. \end{cases}$$

is the clean sample mean-value estimator induced by a trained model $f_\theta$ optimized by maximizing the Evidence Lower Bound (ELBO)

$$\mathcal{L}_{\text{ELBO}}(x^0; \theta) = \sum_{n=1}^{N} \bar{\alpha}_n \mathbb{E}_{x^n \sim q_{n|0}}\left[\delta_{x^n, \text{m}} \cdot (x^0)^\top \log \mu_\theta(x^n, n)\right], \tag{5}$$

where $\delta_{x,y}$ is an indicator function and $q_{n|0} := q(x^n|x^0)$. The ELBO acts as a lower bound for the expected log-likelihood

$$\log p_\theta(x^0) \geq \mathcal{L}_{ELBO}(x^0; \theta). \tag{6}$$

Throughout this work, we abuse the notation such that $x^n$ can be either an integer or its corresponding one-hot vector, whenever it is clear from the context.

# B. Proofs and Derivations

## B.1. Proof to the Forward KL Loss Eq. (FKL Loss)

*Proof.* Denoting the forward KL:

$$
\begin{aligned}
d_{KL}(\pi_{MD}, \pi_\theta; s) &= \sum_{\mathbf{a} \in \mathcal{A}^K} \pi_{MD}(\mathbf{a}|s) \log \frac{\pi_{MD}(\mathbf{a}|s)}{\pi_\theta(\mathbf{a}|s)} \\
&= \sum_{\mathbf{a} \in \mathcal{A}^K} \pi_{MD}(\mathbf{a}|s) \log \pi_{MD}(\mathbf{a}|s) - \sum_{\mathbf{a} \in \mathcal{A}^K} \pi_{MD}(\mathbf{a}|s) \log \pi_\theta(\mathbf{a}|s) \\
&= -\mathcal{H}(\pi_{MD}(\cdot|s)) - \sum_{\mathbf{a} \in \mathcal{A}^K} \pi_{\text{old}}(\mathbf{a}|s) \frac{\exp(q^{\pi_{\text{old}}}(\mathbf{a}, s)/\lambda)}{Z(s)} \log \pi_\theta(\mathbf{a}|s) \\
&\leq -\sum_{\mathbf{a} \in \mathcal{A}^K} \pi_{\text{old}}(\mathbf{a}|s) \frac{\exp(q^{\pi_{\text{old}}}(\mathbf{a}, s)/\lambda)}{Z(s)} \mathcal{L}_{ELBO}(\mathbf{a}, s; \theta) \\
&= \mathbb{E}_{\mathbf{a} \sim \pi_{\text{old}}} \left[ -\frac{\exp(q^{\pi_{\text{old}}}(\mathbf{a}, s)/\lambda)}{Z(s)} \mathcal{L}_{ELBO}(\mathbf{a}, s; \theta) \right],
\end{aligned}
$$

where the first equality is the KL divergence definition, the third equality come from the definition of entropy $\mathcal{H}$ and the definition of the MD policy in Eq. (3), the inequality comes from the ELBO inequality w.r.t $\log \pi_\theta(\mathbf{a}|s)$ of the discrete diffusion policy and from the fact that entropy is has a non-negative value. □

That is, we can bound the forward KL metric using the self-imitation objective. Given the support of $\pi_{\text{old}}$: $\mathcal{A}(\pi_{\text{old}}; s) = \{\mathbf{a}^0 | \pi_{\text{old}}(\mathbf{a}^0|s) > 0, \mathbf{a}^0 \in \mathcal{A}^k\}$, the self-imitation loss is a weighted average of the discrete diffusion loss with softmax weights $w_\lambda$ over this support. The SI loss is a generalized version of the classification policy iteration (Lazaric et al., 2016), which is the solution w.r.t non-regularized MD policy:

**Remark 2** (Classification Discrete Diffusion Loss). *Consider the limit MD policy's temperature $\lambda \to 0$, we get that:*

$$
\lim_{\lambda \to 0} \mathbb{E}_{\mathbf{a} \sim \pi_{\text{old}}} \left[ -\frac{\exp(q^{\pi_{\text{old}}}(\mathbf{a}, s)/\lambda)}{Z(s)} \mathcal{L}_{ELBO}(\mathbf{a}, s; \theta) \right] = -\frac{1}{n^*} \sum_{\mathbf{a}^* \in \mathcal{A}^*(\pi_{\text{old}}; s)} \mathcal{L}_{ELBO}(\mathbf{a}^*, s; \theta),
$$

*where, $\mathcal{A}^*(\pi; s) := \arg\max_{\mathbf{a} \in \mathcal{A}(\pi_{\text{old}}; s)} q^\pi(\mathbf{a}, s)$ and $n^* := |\mathcal{A}^*(\pi; s)|$. This is a classification policy iteration (Lazaric et al., 2016) of the discrete diffusion policy .*

Overall, the self-imitation loss encapsulate a weighted behavioral cloning objective w.r.t MD policy. In practice, computing $w_\lambda$ is non-trivial, as sampling actions from the whole action space may be expensive, especially in the domains consider in this work. Therefore, a practical approach would be to estimate $w_\lambda$ by sampling a subset of actions $\hat{\mathcal{A}}_s \sim \pi_{\text{old}}(\cdot|s)$, where $\hat{\mathcal{A}}_s := \{\mathbf{a}_i\}_{i=1}^M, \mathbf{a}_i \sim \pi_{\text{old}}(\cdot|s)$ and perform a softmax over their $q$-function values, which effectively estimates the normalization factor over the sampled set:

$$
Z(s) \approx \frac{1}{M} \sum_{\mathbf{a} \in \hat{\mathcal{A}}_s} \exp(q^{\pi_{\text{old}}}(s, \mathbf{a})/\lambda) = \frac{\hat{Z}(s)}{M}.
$$

This gives us the next approximated loss:

$$
\mathcal{L}_{FKL}(\theta) = -\mathbb{E}_{s \sim \mathcal{D}, \hat{\mathcal{A}}_s \sim \pi_{\text{old}}} \left[ \sum_{\mathbf{a}^0 \in \hat{\mathcal{A}}_s} \hat{w}_\lambda(s, \mathbf{a}^0) \mathcal{L}_{ELBO}(\mathbf{a}^0, s; \theta) \right],
$$

where $\hat{w}_\lambda(s, \mathbf{a}) := \frac{\exp(q^{\pi_{\text{old}}}(s, \mathbf{a})/\lambda)}{\hat{Z}(s)}$ and $\mathcal{D}$ is the replay buffer.

**B.2. Proof to The Reverse KL Loss Eq. (RKL Loss)**

*Proof.* Denoting the reverse KL:

$$
\begin{aligned}
d_{KL}(\pi_\theta, \pi_{MD}; s) &= \sum_{\mathbf{a} \in \mathcal{A}^K} \pi_\theta(\mathbf{a}|s) \log \frac{\pi_\theta(\mathbf{a}|s)}{\pi_{MD}(\mathbf{a}|s)} \\
&= \sum_{\mathbf{a} \in \mathcal{A}^K} \pi_\theta(\mathbf{a}|s) \log \pi_\theta(\mathbf{a}|s) - \sum_{\mathbf{a} \in \mathcal{A}^K} \pi_\theta(\mathbf{a}|s) \log \pi_{MD}(\mathbf{a}|s) \\
&= \sum_{\mathbf{a} \in \mathcal{A}^K} \pi_\theta(\mathbf{a}|s) \log \pi_\theta(\mathbf{a}|s) - \sum_{\mathbf{a} \in \mathcal{A}^K} \pi_\theta(\mathbf{a}|s) \log \pi_{\text{old}}(\mathbf{a}|s) - \lambda^{-1} \sum_{\mathbf{a} \in \mathcal{A}^K} \pi_\theta(\mathbf{a}|s) q^{\pi_{\text{old}}}(\mathbf{a}, s) + Z(s) \\
&= d_{KL}(\pi_\theta, \pi_{\text{old}}; s) - \lambda^{-1} \mathbb{E}_{\mathbf{a} \sim \pi_\theta}[q^{\pi_{\text{old}}}(\mathbf{a}, s)] + Z(s)
\end{aligned}
$$

Since the normalization factor is independent of $\theta$

$$
\arg\min_\theta d_{KL}(\pi_\theta, \pi_{MD}; s) = \arg\max_\theta \mathbb{E}_{\mathbf{a} \sim \pi_\theta}[q^{\pi_{\text{old}}}(\mathbf{a}, s)] - \lambda d_{KL}(\pi_\theta, \pi_{\text{old}}; s)
$$

which is the mirror decent objective regularized with $\pi_{\text{old}}$. □

# C. Implementation Details

### C.1. RL-D$^2$ Implementation Details for Atari

We follow similar off-policy distributed RL framework as R2D2 (Kapturowski et al., 2018) implemented on ACME (Hoffman et al., 2020). In Atari games, we leverage the same recurrent feature extraction in (Kapturowski et al., 2018) by unrolling an LSTM network. We leverage the priority experience replay (Horgan et al., 2018). The hyperparameters are listed in Table 3.

*Table 3.* RL-D$^2$ Hyperparameters with FKL

| Parameters | Value | Parameters | Value |
|---|---|---|---|
| Number of samples | 1e8 | Sample-to-insert Ratio | 4.0 |
| Number of parallel actors | 16 | Mini Batch size | 64 |
| Unroll length | 40 | Burn-in length | 8 |
| $Q$-network | Transformer | Target update rate | 0.005 |
| Policy network | Transformer | Actor learning rate | $1 \times 10^{-4}$ |
| Tempearture learning rate | $1 \times 10^{-2}$ | Critic leaning rate | $1 \times 10^{-4}$ |
| Transformer hidden dim | 80 | Transformer layers | 3 |
| Transformer heads | 1 | Discounted factor | 0.997 |
| Priority exponent | 0.99 | Replay buffer size | $5 \times 10^6$ |

**Model Architechtures.** The same 3-layer convolutional network structure as (Kapturowski et al., 2018; Mnih et al., 2015) is used for all the algorithms, followed by an LSTM with 512 hidden units, which feeds into an actor and value networks implemetned as transformers.

The learnable parameters inside our transformer include the input embedding, linear projections for conditioning, weights/biases in the multi-head attention and feed-forward networks within each transformer block. Key learnable parameters are also the adaptive normalization layers (lns) that generate dynamic shift, scale, and gate values based on the conditioning. Finally, the output projection is learnable. Conditioning is introduced via a FiLM-like (Feature-wise Linear Modulation) mechanism. The objective then passes through small linear networks to produce shift, scale, and gate parameters. These dynamically modulate activations after layer normalization in both the attention and feed-forward sub-layers. For example, inputs to sub-layers become `norm(h) * (scale + 1.0) + shift`, with gate controlling residual connections. This enables the transformer to adapt its internal computations layer-wise based on external conditions.

## C.2. Temperature Tuning By Hard KL Divergence Constraints

Autotuning the temperature parameters $\lambda$ has been a challenging problem for RL with diffusion policies, as the output log probabilities are unknown. Existing methods leverage Gaussian mixture fitting (Wang et al., 2024b), uniform data insertion (Ding et al., 2024), and data processing inequalities (Celik et al., 2025).

We noticed a simple approach using duality by enforcing a hard constraint on the KL divergence based on Abdolmaleki et al. (2018). Consider the policy mirror descent with hard constraints,

$$\max_{\pi} \mathbb{E}_{\mathbf{a} \sim \pi}[A^{\pi_{\mathrm{old}}}(s, \mathbf{a})]$$
$$\text{s.t. } d_{\mathrm{KL}}(\pi, \pi_{\mathrm{old}}) \leq \epsilon \tag{7}$$

To solve it, we construct the Lagrangian

$$L(\pi, \lambda, \eta) = \mathbb{E}_{a \sim \pi}[A^{\pi_{\mathrm{old}}}(s, \mathbf{a})] + \lambda(\epsilon - d_{\mathrm{KL}}(\pi, \pi_{\mathrm{old}})) + \eta(1 - \sum_a \pi(\mathbf{a}|s))$$

Gradient to the primal objective,

$$\partial \pi L = A^{\pi_{\mathrm{old}}}(s, \mathbf{a}) - \lambda(\log \frac{\pi(\mathbf{a}|s)}{\pi_{\mathrm{old}}(a|s)} + 1) + \eta$$

Let it equal $0$ we get the primal optimal solution is

$$\pi = \pi_{\mathrm{old}}(a|s) \exp(\frac{A^{\pi_{\mathrm{old}}}(s, \mathbf{a})}{\lambda}) \exp(\frac{\eta - \lambda}{\lambda})$$

As we have the normalization constraint, we have

$$\exp(-\frac{\eta - \lambda}{\lambda}) = \sum_{\mathbf{a}} \pi_{\mathrm{old}}(a|s) \exp(\frac{A^{\pi_{\mathrm{old}}}(s, \mathbf{a})}{\lambda}) := Z$$

Therefore, we have

$$\eta = \lambda(1 - \log Z)$$

Substituting this back to the Lagrangian, we have

$$
\begin{aligned}
g(\lambda) =& \lambda\epsilon + \eta + \sum_{\mathbf{a}} \pi(\mathbf{a}|s) \left(A^{\pi_{\mathrm{old}}}(s, \mathbf{a}) - \lambda \log\left(\exp(A^{\pi_{\mathrm{old}}}(s, \mathbf{a})/\lambda)\right) + \lambda \log Z - \eta\right) \\
=& \lambda\epsilon + \sum_{\mathbf{a}} \pi(\mathbf{a}|s)(\lambda \log Z) \\
=& \lambda\epsilon + \lambda \log Z \\
=& \lambda\epsilon + \lambda \log \sum_{\mathbf{a}} \pi_{\mathrm{old}}(a|s) \exp(\frac{A^{\pi_{\mathrm{old}}}(s, \mathbf{a})}{\lambda}) \\
\approx& \lambda\epsilon + \lambda \log \frac{1}{N} \sum_{i=1}^{N} \exp(\frac{A^{\pi_{\mathrm{old}}}(s, \mathbf{a}_i)}{\lambda}) \qquad\qquad a_i \sim \pi_{\mathrm{old}}(\mathbf{a}_i|s) \\
=& \lambda\epsilon + \lambda \operatorname{logsumexp}(\frac{A^{\pi_{\mathrm{old}}}(s, \mathbf{a}_i)}{\lambda}) - \lambda \log N
\end{aligned}
$$

Therefore, we can update the temperature parameter by $\min g(\lambda)$, which can be used in discrete and continuous diffusion.

## C.3. Sampling Techniques.

- **Top-p sampling.** For each action that is sampled to be unmasked, we select the smallest set of actions whose cumulative probability computed from $f_\theta$ exceeds $P = 0.98$. Then we re-normalize the distribution including only these actions and sample from this re-normalized distribution.

- **Re-masking.** Using the same techniques in (Wang et al., 2025), we don't need to change the ELBO Eq. (5) as well as the FKL policy loss Eq. (FKL Loss). We only need to change

# D. Experiments

## D.1. DNA Generation Setup

**Dataset.** The experiment is based on a large, publicly available dataset of enhancers, which contains activity measurements for approximately 700,000 DNA sequences, each 200 base pairs long, within human cell lines. The dataset contains the expressive level, which is also used to train our reward models. A masked discrete diffusion model was pretrained on the complete set of sequences.

**Reward models.** Following established conventions in (Wang et al., 2024a), the dataset was then divided by chromosome to train two distinct reward models, or "oracles". These oracles, built on the Enformer (Avsec et al., 2021) architecture, were designed to predict the enhancer activity level in the HepG2 cell line; one oracle was used to fine-tune the models, while the other was reserved for evaluation.

**Evaluation Metrics.** To conduct a thorough assessment of each model's ability to generate effective enhancers, the following metrics were employed:

1. Predicted Activity (Reward): This metric measures the enhancer activity level in the HepG2 cell line as predicted by the evaluation reward oracle. It's important to note that the models were fine-tuned using a separate oracle trained on a different chromosomal subset of the data.

2. Approximated Log-Likelihood (App-Log-Lik): The log-likelihood of the generated sequences was calculated with respect to the pretrained model. This measures how "natural" the sequences are; a low likelihood would indicate that the model over-optimized the reward and generated out-of-distribution sequences.

## D.2. Implementation of the Macro baselines

We conducted the following algorithm that converts baseline algorithms to the setup with macro actions.

- **DQN-Macro:** For DQN, we directly make the $Q-$network output to be in the shape of $|\mathcal{A}|^K$, which is the size of the total combinatorial action space.

- **IMPALA-Macro:** Instead of output $|\mathcal{A}|$ logits for a single action, the actor networks predict $K \times |\mathcal{A}|$, where each $|\mathcal{A}|$-dim vector is the logits for one action in the macro actions.

### D.2.1. HYPERPARAMETER SEARCH FOR THE DQN-MACRO

We conduct hyperparameter search for the baseline **DQN-Macro** algorithm to see that whether the our macro baselines prefer different hyper parameters from the original algorithm. we conduct search on the following 3 parameters:

- MLP layers. The original paper include a 1-layer MLP head over the deep residual feature extractors. We increase the number of layers to **2 and 3** to see whether the increase number of parameters benefit macro actions.

- Prioritized experience replay exponents. Current one used for DQN is 0.6 and we try 0.4.

- The value of $\epsilon$ in the $\epsilon$-greedy exploration. Current one used for DQN is 0.01 and we try 0.001 and 0.1.

The results are shown in Fig. 7, which shows that the default DQN parameters, also the one we used in our main results, still perform the best for the DQN with macro actions.

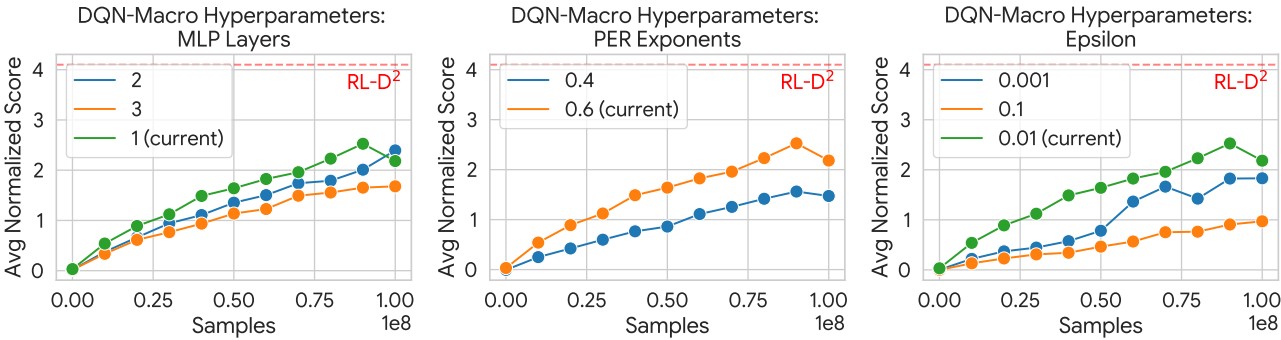

*Figure 7.* DQN-Macro hyperparameter search. The results shows averaged human normalized score over 10 Atari environments, each with 3 random seeds. Red dash line is the proposed RL-D$^2$ for reference.

### D.3. Scalability and Horizon-Complexity Trade-off Setup

Compared to the hyper parameters in Table 3, we scale the computation by the following conditions:

- **Number of environment steps:** $5 \times 10^8$ maximum for macro action length 8 and 16.

- **Model sizes:** We change the heads of the transformers, 2 heads for macro aciton 4, 4 heads for macro aciton 8, 6 heads for macro aciton length 16.

- **Batch size.** We change the mini-batch size to 128, resulting in a total effective batch size of 4096 for macro action length 8 and 16.

### D.4. RL-D$^2$ Implementation of Google Research Football

For the Google Research Football we used the same features using in (Song et al., 2023) and the same scenarios settings for training and evaluation. We implemented the RKL version of RL-D$^2$ with a single-step ratio in a PPO (Schulman et al., 2017) framework. For the state embeddings we used an additional transformer similar to the one used for the diffusion process. The transformer outputs an state embedding for each player, which are fed as a condition for the action-transformer while also fed to a value-head MLP network that outputs a value for each player. The value is trained as mentioned in (Wen et al., 2022a). We trained the model and baselines over 1G enviornment steps for *11 vs 11* and 500M for the rest of the scenarios.

*Table 4.* Google Research Football Hyperparameters

| Parameters | Value | Parameters | Value |
|---|---|---|---|
| Critic LR | 5e-4 | Sample-to-insert Ratio | 1.0 |
| Actor LR | 5e-4 | Batch size | 256 |
| Discount factor | 0.995 | Number of mini-batch | 1 |
| Number of actors | 256 | Max grad norm | 0.5 |
| Entropy coeff | 1e-2 | Discount factor | 0.995 |
| Training epochs | 10 | Rollout size | 1024 |
| Diffusion steps | Num of players | Ratio clip | 0.2 |

### D.5. Computation of the Stratified Bootstrap Confidence Interval

To compute the stratified bootstrap confidence interval for aggregated Atari metrics, we use a resampling process designed to provide a statistically rigorous characterization of algorithm performance across high-variance environments. The computation involves performing 50,000 bootstrap iterations to ensure stability in the resulting estimates. In each iteration, a synthetic dataset of 168 runs is generated by **sampling with replacement** from the original pool of 3 seeds for each of the 56 tasks (in total 168 tasks). This "stratified" approach means that the resampling occurs independently within each

environment (the strata), ensuring that each game contributes a consistent weight to the aggregate metric across every iteration.

Specifically, let $X_{t,s}$ be the human-normalized score for task $t \in \{1, \ldots, 56\}$ and seed $s \in \{1, \ldots, 3\}$. The total number of runs is $M = 56 \times 3 = 168$. In each of the $B = 50,000$ bootstrap iterations, we perform stratified resampling:

$$\mathcal{X}_b^* = \bigcup_{t=1}^{56} \{X_{t,1}^*, \ldots, X_{t,3}^*\}_b$$

where each $X_{t,s}^*$ is sampled with replacement from the original set of seeds $\{X_{t,1}, X_{t,2}, X_{t,3}\}$ for that specific task. This stratification ensures that every bootstrap sample $\mathcal{X}_b^*$ maintains exactly 3 runs per environment. We then calculate the aggregate statistic, such as the Mean ($\mu_b^*$), Median, or Interquartile Mean (IQM):

$$\mu_b^* = \frac{1}{168} \sum_{i=1}^{168} x_i^*, \quad x_i^* \in \mathcal{X}_b^*$$

The IQM is calculated by discarding the lowest and highest 25% of the $M$ scores in the resampled set and averaging the remaining 50%. The final 95% confidence interval is determined by the $2.5^{th}$ and $97.5^{th}$ percentiles of the empirical distribution formed by these $B$ bootstrap statistics:

$$[q_{0.025}(\text{stat}_1^*, \ldots, \text{stat}_B^*), \quad q_{0.975}(\text{stat}_1^*, \ldots, \text{stat}_B^*)]$$

This method allows the researchers to provide the robust intervals seen in Table 6 and Figure 8, which account for the high variance and non-normal distribution typical of reinforcement learning benchmarks.

For every one of the 50,000 resampled datasets, an aggregate statistic—such as the mean, median, or Interquartile Mean (IQM) of human-normalized scores—is calculated. The final 95% confidence interval is derived from the distribution of these calculated statistics, offering a robust measure of uncertainty that, unlike standard error, does not rely on the assumption of a normal performance distribution.

## E. Additional Experimental Results

### E.1. More Atari Aggregated Results

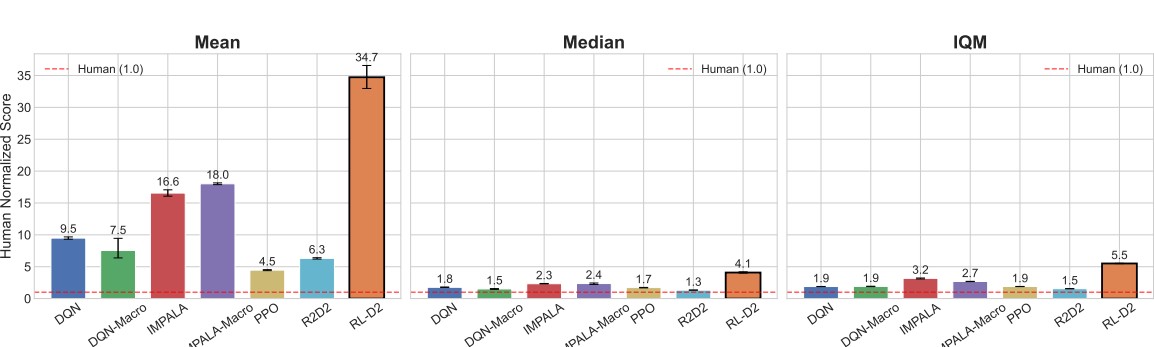

*Figure 8.* Statistics of Human-Normalized Atari scores. 95% Confidence intervals are computed with stratified bootstrap sampling of 50,000 bootstrap iterations from 168 runs (3 seeds * 56 tasks).

*Table 6.* Detailed numbers of the statistics and confidence intervals. Confidence intervals are computed with stratified bootstrap sampling of 50,000 bootstrap iterations.

| | Mean | Median | IQM |
|---|---|---|---|
| DQN | 9.49 [ 9.31, 9.68] | 1.77 [ 1.76, 1.81] | 1.89 [ 1.88, 1.90] |
| DQN-Macro | 7.54 [ 6.38, 9.46] | 1.51 [ 1.44, 1.57] | 1.91 [ 1.88, 1.94] |
| IMPALA | 16.55 [16.07, 17.07] | 2.33 [ 2.30, 2.37] | 3.17 [ 3.08, 3.23] |
| IMPALA-Macro | 18.01 [17.88, 18.15] | 2.36 [ 2.22, 2.45] | 2.69 [ 2.66, 2.71] |
| PPO | 4.47 [ 4.39, 4.54] | 1.73 [ 1.68, 1.76] | 1.90 [ 1.88, 1.91] |
| R2D2 | 6.31 [ 6.20, 6.41] | 1.32 [ 1.32, 1.34] | 1.55 [ 1.54, 1.56] |
| RL-D$^2$ | **31.33 [29.74, 33.00]** | **4.09 [ 4.00, 4.20]** | **4.98 [ 4.96, 5.00]** |

Performance Difference between RL-D$^2$ and Baselines, 95 % CIs are computed by stratified bootstrap sampling with 50k iterations

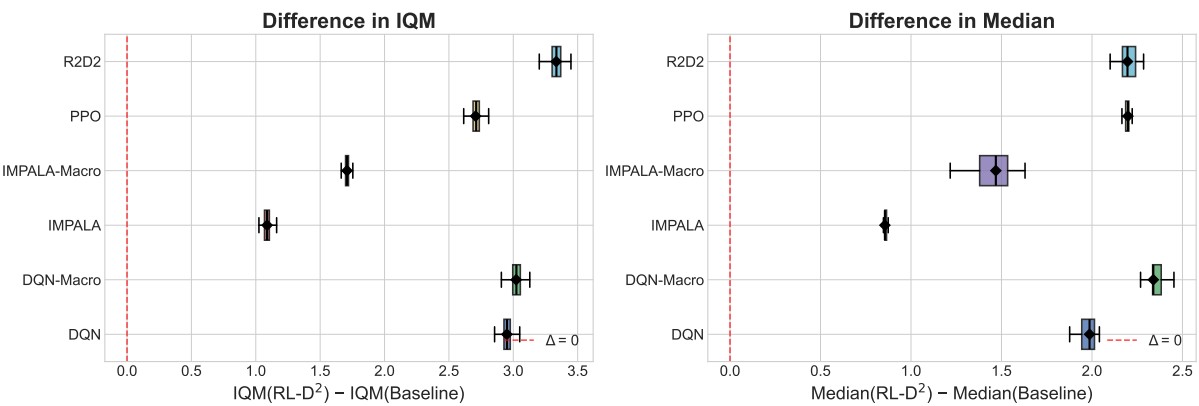

*Figure 9.* Performance difference statistics. 95% CIs are computed by stratified bootstrap sampling with 50k iterations from 168 runs (3 seeds * 56 tasks).

### E.2. Full Results for Atari Games

Please refer to Table 5 for the full results of Atari games, and Fig. 10 for the comparison with the best baselines. We outperm all the baselines in 36 out of 56 Atari environments.

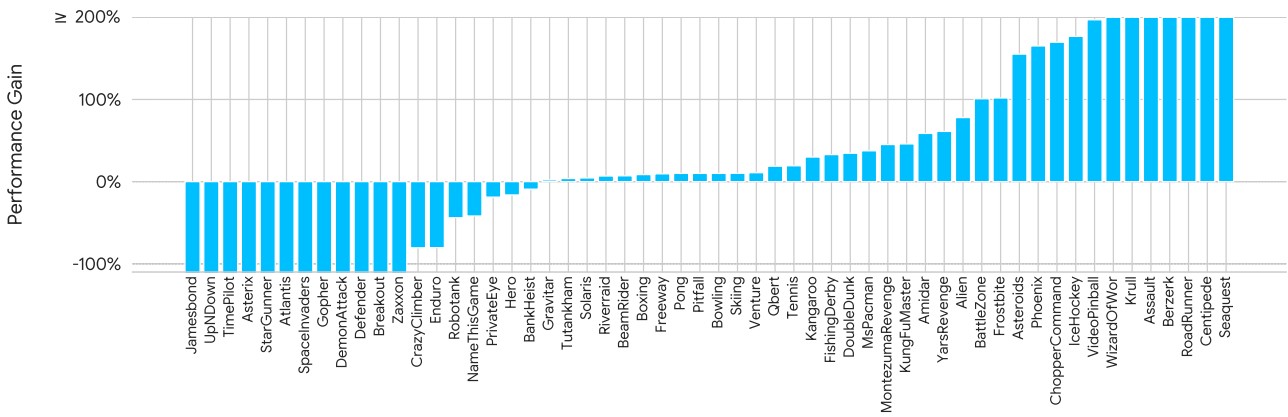

*Figure 10.* Mean human normalized score of RL-D$^2$ compared to the best baselines in each Atari task.

### E.3. Ablation of temperature tuning

We conduct ablation studies on the temperature tuning discussed in Appendix C.2 on multiple benchmarks including MinAtar, Atari and Google Football.

#### E.3.1. MINATAR

For MinAtar, the current temperature is updated following a KL constraint is set to 1.0. We compare auto-tuning with fixed temperatures. The results are shown in Fig. 11, showing the auto-tuning consistently outperforms fixed temperature.

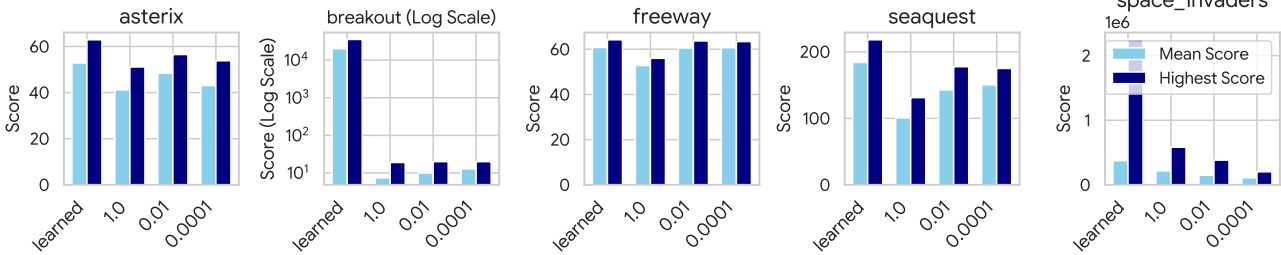

*Figure 11.* Ablation studies of temperature tuning with fixed temperature variable. Bars indicates the mean episode returns over last 100 evaluations over 3 seeds.

#### E.3.2. ATARI GAMES

For Atari games, we use KL constraint schedule linearly decaying from 1 to 0.1 in the first $10^7$ samples. The intuition behind this selection is that, in vast combinatorial discrete spaces, a larger initial KL constraint allows the policy to deviate significantly from initialization, enabling the broad exploration necessary to discover high-value actions.

We compare auto-tuning with other two temperature control scheme: Fixed KL constraints 1.0 and 0.1; The results are shown in Fig. 12, showing that smaller KL constraints like 0.1 fails to learn in the initial phase. Large KL constraints like 1.0 will cause instabilities after 50M steps, which makes the performance worse. The linear decay achieves initial fast learning and overall stability.

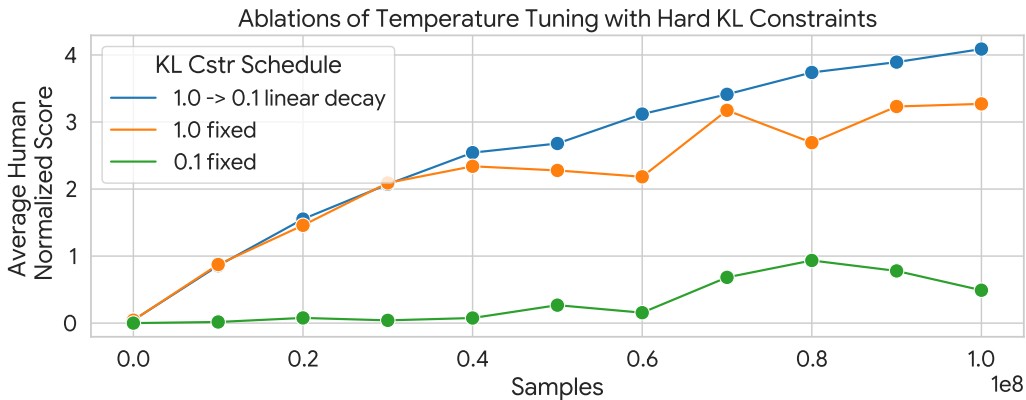

*Figure 12.* Ablation studies of temperature tuning with hard KL constraints, compared with different KL constraint schedules. The results shows averaged human normalized score over 10 Atari environments, each with 3 random seeds.

#### E.3.3. GOOGLE FOOTBALL

We compare three different KL constraint schedule in Google Football, 0.1, 1.0, and 10. the results are shown in Fig. 13, which shows another failure mode of collapsing for larger KL constraints. The reason might be that larger KL constraints push the policy to be highly deterministic. Therefore, the collected data lost diversity, making the training failed.

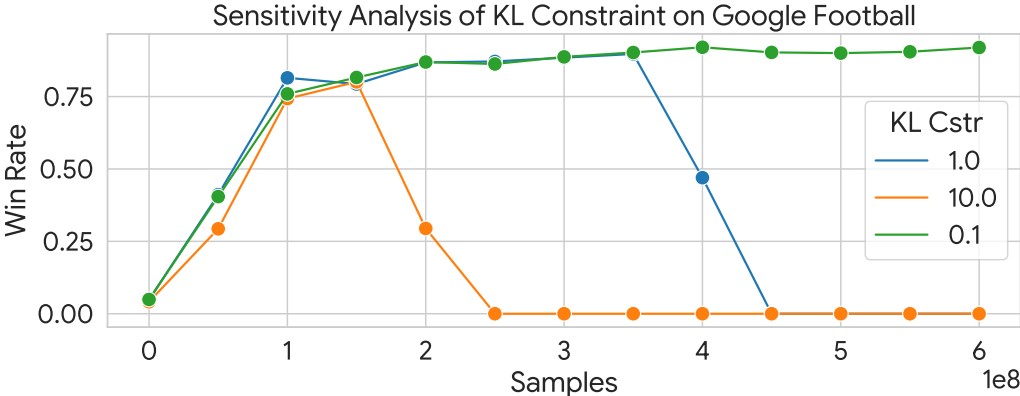

*Figure 13.* Sensitivity studies of temperature tuning with hard KL constraints on Google Football 11 vs 11. The results shows averaged win rate over 5 random seeds.

In summary, the temperature tuning is significant to the performance of RL-D$^2$-FKL. When selecting the KL constraints, we should consider two key factors: (1) The initial KL constraint should be large so that the policy can start learning; (2) Avoid collapsing in the later training phase by enforcing not too large KL constraints. Although not necessary, a decay schedule is very helpful to stabilize the training and get better performance.

### E.4. Discrete Diffusion as Planner for Causal Action Spaces

In applications of macro actions in Atari games, we can just commit to the first action rather than all the macro actions. Therefore, it is common to plan for a longer trajectory and only commits to the first action, such as model predictive control and Monte-Carlo tree search (Garcia et al., 1989; Silver et al., 2016). However, if we would like to implement planning in online RL, the parameterization of the planning trajectory is not trivial. If we use autoregressive models as our the parameterization to generate the trajectory, the next action will depend only on the current state and not depend on the planned trajectory, leaving future actions useless.

Benefiting from the non-casual unmasking of discrete diffusion models, we can directly use the discrete diffusion to parameterize the planner. **Note that this is a total different setup and algorithm from the main text Sec. 5.2**. We show the planner performance with the same set of hyperparameters in Table 3 with varying planning steps shown in Fig. 14. The performance increase with increasing planning length, showcasing that only committing to the first action increase the robustness of committing all macro actions. The planner is also scalable with respect to the increasing size of action spaces.

---

**Algorithm 1** Discrete Diffusion as Planners

---

**Require:** Planning length $K$, current policy $\pi_{\text{old}}$, replay buffer $\mathcal{D}$, current value function $q^{\pi_{\text{old}}}$.
1: # Policy updates in training.
2: For states $s \sim \mathcal{D}$, sample macro actions $\mathbf{a} = (a_1, \ldots, a_k)$, compute $\mathcal{L}_{\text{ELBO}}$ with Eq. (5).
3: Take the first one to compute value function $q^{\pi_{\text{old}}}(s, a_1)$.
4: Optimize the policy by self-imitation loss Eq. (FKL Loss).
5: # Inference.
6: Sample macro actions $(a_1, \ldots, a_k)$ and only take $a = a_0$.

---

### E.5. Ablation of On-Policy Diffusion Training.

We conduct ablation studies on the on-policy training discussed in Sec. 4.2 and the results are shown in Fig. 15, which shown the on-policy diffusion training help improve the performance.

### E.6. Ablation of Network Architecture.

We compare using multi-layer perceptrons (MLP) versus transformers (Vaswani et al., 2017) for parameterization of the $Q$ and policy networks. The two networks share the same amount of parameters around $4 \times 10^5$. We can see the transformer

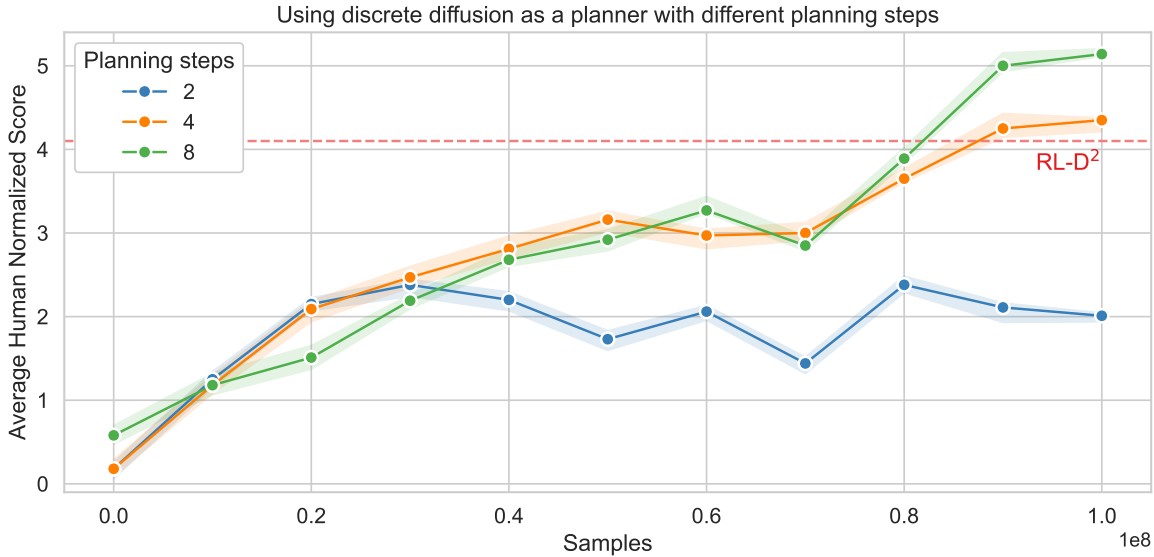

*Figure 14.* Performance with using discrete diffusion as a planner with different length of planning steps, averaged over 10 Atari games with 3 random seed each game.

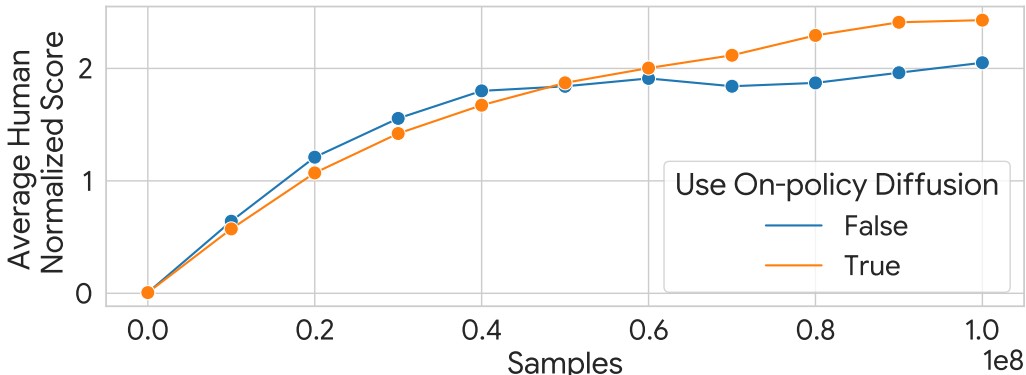

*Figure 15.* Ablation studies of on-policy diffusion training. The curves indicates mean reward using macro length 8 over 10 representative Atari environments, 3 seed, and 10 consecutive evaluation episodes.

consistently outperform MLP.

## F. Baseline Selection Protocol

Our baseline selection protocol was designed to rigorously evaluate the performance, scalability, and versatility of $RL - D^2$ across the distinct challenges presented by combinatorial action spaces. We selected a diverse set of baselines, ranging from established standards in reinforcement learning to domain-specific state-of-the-art methods, ensuring that our comparisons were fair, comprehensive, and directly addressed the central claims of our work.

The selection was tailored to the three specific experimental domains:

### F.1. DNA Sequence Generation

This domain tests the single-step policy optimization (combinatorial bandit) capabilities of our framework. The baselines were chosen to cover two main categories:

- **Controlled Generation Methods:** We included standard guidance-based techniques (CG, SMC, TDS, and CFG).

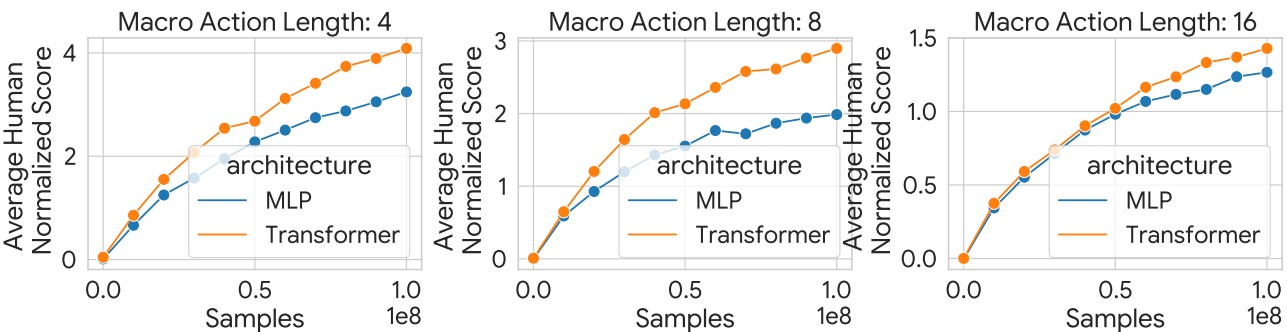

*Figure 16.* Ablation studies of on-policy diffusion training. The curves indicates mean reward using macro length 8 over 10 representative Atari environments, 3 seed, and 10 consecutive evaluation episodes.

These methods are not based on RL policy optimization but are common for guiding generative models toward desired properties. This comparison validates RL-D$^2$ against non-RL finetuning approaches.

- **State-of-the-Art RL Finetuning:** We included **DRAKES**, a strong, recent baseline that also uses RL to optimize a generative model for sequence generation. DRAKES employs the Gumbel-Softmax trick to enable reward back-propagation through the generative process. This provides a direct comparison against another RL-based approach for finetuning discrete generative models.

### F.1.1. MACRO ACTIONS

This domain tests the ability of RL-D$^2$ to handle online RL in complex, long-horizon tasks by modeling sequences of primitive actions.

- **Standard RL Baselines:** We first included strong, general-purpose RL algorithms (DQN, IMPALA, PPO, R2D2) that operate on a single-action level. These baselines establish a performance reference and demonstrate the inherent difficulty of the tasks without temporal abstraction.

- **Adapted Macro-Action Baselines:** To create a direct comparison, we adapted standard algorithms to handle macro-actions, as detailed in **Appendix D.2**:

    - **DQN-Macro:** This baseline represents a "naive" approach, where the Q-network's output layer is expanded to $|\mathcal{A}|^K$. This directly exposes the challenge of exponential scaling that RL-D$^2$ is designed to overcome.
    - **IMPALA-Macro:** This baseline represents a "factored" approach, where the policy network outputs $K \times |\mathcal{A}|$ logits, assuming conditional independence between actions in the macro-action sequence.

These adaptations allow us to test our hypothesis that an expressive, non-causal generative model (our diffusion policy) can outperform both naive exponential-space methods and simple independent factorization methods.

### F.2. Cooperative Multi-Agent Reinforcement Learning

This domain tests RL-D$^2$ on modeling the combinatorial *joint action* of multiple agents.

- **Autoregressive (AR) Policy:** We selected a strong **autoregressive transformer** policy as our primary baseline. This is a dominant and highly effective paradigm for modeling joint actions, where the action for each agent is sampled sequentially, conditioned on the actions of previous agents.

- This comparison is central to our motivation. The introduction (Section 1) explicitly notes that AR models impose an artificial causal ordering. By comparing RL-D$^2$ (a non-causal generative model) against a strong AR baseline, we directly test our claim that diffusion's flexible, non-causal generation process is a superior parameterization for modeling complex inter-agent dependencies in MARL.

# G. Additional Discussions.

**Compared to discrete black-box solvers.** Compared with standard discrete black-box solvers, our RL-D$^2$ framework is particularly well suited to sequential decision-making problems with large combinatorial action spaces. While methods such as evolutionary algorithms and zeroth-order optimization typically treat each problem instance as a static objective and often optimize from scratch, RL-$D^2$ learns a state-conditioned diffusion policy that can exploit transition dynamics, perform structured exploration, and generalize to unseen states without re-solving each instance independently. The expressive diffusion policy also provides a scalable distributional model over high-dimensional discrete actions, mitigating some of the curse-of-dimensionality issues faced by generic black-box search methods. In addition, our FKL variant supports off-policy learning, enabling reuse of previously collected trajectories or offline datasets, which can further improve query efficiency when environment interactions are expensive.

| Tasks | DQN | DQN-Macro | IMPALA | IMPALA-Macro | PPO | R2D2 | RL-D$^2$ |
|---|---|---|---|---|---|---|---|
| Alien | 11.49k ± 105.46 | 2.63k ± 43.32 | 9.04k ± 201.70 | 12.40k ± 183.48 | 17.84k ± 480.29 | 8.09k ± 89.89 | 22.53k ± 848.31 |
| Amidar | 3.43k ± 37.04 | 691.56 ± 29.63 | 3.06k ± 12.48 | 5.60k ± 39.90 | 1.44k ± 1.89 | 1.64k ± 35.44 | 6.44k ± 71.30 |
| Assault | 5.92k ± 137.76 | 6.46k ± 433.42 | 18.34k ± 884.87 | 18.29k ± 467.93 | 5.23k ± 121.56 | 3.50k ± 96.09 | 20.99k ± 286.49 |
| Asterix | 4.56k ± 73.22 | 24.42k ± 611.26 | 361.80k ± 6.94k | 28.07k ± 563.07 | 37.57k ± 1.92k | 4.67k ± 64.12 | 133.56k ± 7.46k |
| Asteroids | 1.51k ± 23.60 | 2.11k ± 15.32 | 5.71k ± 112.65 | 8.95k ± 145.30 | 14.49k ± 266.01 | 2.01k ± 32.60 | 82.24k ± 5.90k |
| Atlantis | 984.11k ± 13.14k | 802.66k ± 50.57k | 1030.98k ± 6.81k | 864.89k ± 1.24k | 710.85k ± 18.11k | 1082.26k ± 40.63k | 1003.63k ± 1.79k |
| BankHeist | 1.84k ± 18.02 | 1.25k ± 19.15 | 1.50k ± 2.70 | 1.10k ± 4.31 | 485.20 ± 4.21 | 944.20 ± 3.53 | 1.70k ± 6.59 |
| BattleZone | 116.15k ± 1.48k | 43.66k ± 837.10 | 68.43k ± 1.12k | 166.30k ± 2.96k | 54.66k ± 472.35 | 76.09k ± 1.69k | 197.94k ± 3.08k |
| BeamRider | 3.55k ± 99.35 | 5.62k ± 815.84 | 21.07k ± 837.41 | 14.82k ± 143.88 | 29.47k ± 774.52 | 3.10k ± 78.03 | 28.98k ± 1.10k |
| Berzerk | 340.95k ± 24.42k | 1.11k ± 19.22 | 1.49k ± 45.20 | 7.43k ± 269.77 | 1.29k ± 33.72 | 110.48k ± 42.94 | 802.99k ± 64.87k |
| Bowling | 266.38 ± 0.01 | 41.94 ± 6.70 | 70.00 ± 0.00 | 54.80 ± 0.13 | 149.03 ± 0.25 | 197.18 ± 0.17 | 266.38 ± 0.00 |
| Boxing | 97.13 ± 0.20 | 99.31 ± 0.10 | 100.00 ± 0.00 | 98.60 ± 0.02 | 98.99 ± 0.11 | 97.89 ± 0.12 | 99.51 ± 0.00 |
| Breakout | 76.39 ± 2.94 | 401.10 ± 1.61 | 675.48 ± 18.21 | 161.19 ± 5.48 | 394.17 ± 0.69 | 124.08 ± 1.44 | 424.15 ± 0.00 |
| Centipede | 36.42k ± 541.25 | 6.26k ± 214.74 | 8.08k ± 381.00 | 27.66k ± 835.60 | 27.60k ± 380.41 | 20.66k ± 247.27 | 68.41k ± 910.77 |
| ChopperCommand | 13.18k ± 320.75 | 3.59k ± 300.73 | 23.86k ± 630.67 | 15.74k ± 709.66 | 2.35k ± 115.21 | 2.52k ± 148.92 | 34.36k ± 609.01 |
| CrazyClimber | 93.75k ± 1.37k | 136.37k ± 3.07k | 136.05k ± 949.62 | 107.05k ± 1.55k | 70.01k ± 1.19k | 118.39k ± 1.04k | 113.69k ± 759.99 |
| Defender | 17.60k ± 256.97 | 51.94k ± 496.01 | 427.49k ± 23.25k | 33.85k ± 618.50 | 55.75k ± 606.50 | 38.70k ± 961.61 | 138.16k ± 7.10k |
| DemonAttack | 3.89k ± 61.93 | 26.78k ± 5.85k | 132.40k ± 126.41 | 44.64k ± 1.58k | 9.91k ± 313.53 | 2.89k ± 35.35 | 55.93k ± 1.34k |
| DoubleDunk | -0.36 ± 0.06 | -3.18 ± 2.10 | 23.46 ± 0.07 | 0.00 ± 0.00 | 5.60 ± 0.36 | -0.86 ± 0.34 | 24.00 ± 0.00 |
| Enduro | 651.59 ± 8.84 | 2.22k ± 58.92 | 8.16 ± 0.49 | 1.15k ± 19.29 | 1.64k ± 48.10 | 852.35 ± 29.13 | 1.45k ± 50.17 |
| FishingDerby | 68.54 ± 0.86 | 26.46 ± 0.52 | 44.99 ± 0.66 | 45.31 ± 0.68 | 0.20 ± 0.67 | 20.74 ± 0.99 | 80.64 ± 0.00 |
| Freeway | 33.82 ± 0.00 | 32.61 ± 0.07 | 32.73 ± 0.03 | 33.46 ± 0.03 | 32.68 ± 0.06 | 34.00 ± 0.00 | 33.84 ± 0.00 |
| Frostbite | 9.41k ± 14.63 | 3.93k ± 194.22 | 862.00 ± 48.06 | 9.00k ± 2.64 | 3.97k ± 429.59 | 10.47k ± 78.34 | 14.39k ± 143.25 |
| Gopher | 2.72k ± 120.47 | 27.47k ± 3.09k | 89.58k ± 3.91k | 27.69k ± 626.48 | 5.38k ± 132.20 | 3.95k ± 274.47 | 67.92k ± 1.32k |
| Gravitar | 2.68k ± 11.18 | 1.10k ± 19.89 | 4.27k ± 2.64 | 4.87k ± 28.31 | 4.57k ± 5.56 | 3.08k ± 28.87 | 4.62k ± 13.83 |
| Hero | 22.90k ± 17.41 | 10.69k ± 452.97 | 28.98k ± 5.71 | 36.74k ± 7.49 | 14.08k ± 10.82 | 32.41k ± 685.43 | 28.94k ± 14.34 |
| IceHockey | 12.45 ± 0.70 | 4.62 ± 0.67 | 26.42 ± 0.15 | 25.69 ± 0.20 | 15.98 ± 0.45 | -0.15 ± 0.16 | 46.60 ± 0.14 |
| Jamesbond | 1.41k ± 50.94 | 584.00 ± 9.20 | 1.74k ± 66.32 | 75.06k ± 80.92 | 1.60k ± 155.90 | 1.08k ± 18.32 | 18.21k ± 3.59k |
| Kangaroo | 14.67k ± 113.45 | 8.96k ± 273.42 | 14.50k ± 0.00 | 14.22k ± 8.61 | 2.00k ± 0.00 | 12.80k ± 127.16 | 15.26k ± 10.64 |
| Krull | 69.80k ± 1.86k | 9.36k ± 132.50 | 10.02k ± 38.49 | 83.59k ± 2.56k | 8.97k ± 83.18 | 54.57k ± 1.02k | 385.51k ± 3.70k |
| KungFuMaster | 26.71k ± 236.87 | 36.65k ± 718.07 | 55.69k ± 1.09k | 15.98k ± 206.24 | 31.15k ± 465.11 | 54.57k ± 1.02k | 63.76k ± 789.60 |
| MontezumaRevenge | 835.28 ± 36.72 | 0.00 ± 0.00 | 0.00 ± 0.00 | 0.00 ± 0.00 | 400.00 ± 0.00 | 400.00 ± 0.00 | 2.50k ± 7.26 |
| MsPacman | 20.18k ± 162.54 | 3.90k ± 130.60 | 8.85k ± 77.01 | 7.77k ± 63.70 | 21.04k ± 250.98 | 9.82k ± 68.46 | 22.87k ± 3.39 |
| NameThisGame | 6.86k ± 68.39 | 14.91k ± 442.47 | 15.64k ± 90.75 | 13.69k ± 109.46 | 8.40k ± 68.86 | 7.15k ± 81.37 | 12.66k ± 197.84 |
| Phoenix | 4.98k ± 66.88 | 14.11k ± 1.41k | 192.34k ± 11.13k | 6.37k ± 38.35 | 43.91k ± 3.04k | 5.61k ± 46.40 | 202.39k ± 5.33k |
| Pitfall | 0.00 ± 0.00 | -2.48 ± 0.94 | 0.00 ± 0.00 | 0.00 ± 0.00 | 0.00 ± 0.00 | 0.00 ± 0.00 | 0.00 ± 0.00 |
| Pong | 13.31 ± 0.12 | 19.70 ± 0.13 | 20.47 ± 0.09 | 12.25 ± 0.22 | 21.00 ± 0.00 | 19.73 ± 0.11 | 21.00 ± 0.00 |
| PrivateEye | 35.22k ± 4.16 | 100.00 ± 0.00 | 389.34 ± 268.73 | 99.64 ± 0.02 | 1.59k ± 0.57 | 35.12k ± 28.48 | 15.18k ± 4.02 |
| Qbert | 29.68k ± 62.59 | 9.00k ± 431.38 | 15.66k ± 386.65 | 475.04 ± 0.03 | 5.49k ± 263.05 | 16.68k ± 213.54 | 30.85k ± 57.30 |
| Riverraid | 7.48k ± 81.04 | 17.07k ± 492.67 | 18.44k ± 100.97 | 9.87k ± 39.11 | 8.01k ± 56.74 | 11.33k ± 90.61 | 17.94k ± 98.91 |
| RoadRunner | 317.32k ± 9.62k | 52.61k ± 630.59 | 59.20k ± 368.83 | 514.91k ± 7.59k | 23.88k ± 212.10 | 77.74k ± 1.95k | 563.86k ± 2.53k |
| Robotank | 33.33 ± 0.72 | 66.72 ± 0.49 | 71.97 ± 0.33 | 66.67 ± 0.17 | 63.62 ± 0.51 | 31.22 ± 0.20 | 66.74 ± 0.79 |
| Seaquest | 3.97k ± 37.56 | 24.67k ± 5.63k | 27.12k ± 824.83 | 10.85k ± 134.19 | 6.87k ± 72.78 | 3.63k ± 194.09 | 144.62k ± 4.65k |
| Skiing | -4.43k ± 5.26 | -29.41k ± 266.48 | -9.01k ± 0.07 | -8.95k ± 0.00 | -15.00k ± 111.88 | -27.93k ± 185.34 | -4.41k ± 7.47 |
| Solaris | 15.00k ± 517.51 | 3.26k ± 181.94 | 2.22k ± 113.16 | 2.41k ± 47.98 | 6.00k ± 146.73 | 3.02k ± 160.32 | 14.39k ± 263.47 |
| SpaceInvaders | 2.02k ± 30.80 | 6.22k ± 714.13 | 41.54k ± 8.75k | 9.92k ± 329.91 | 3.87k ± 56.20 | 1.75k ± 59.99 | 10.86k ± 480.94 |
| StarGunner | 1.40k ± 22.05 | 96.97k ± 7.53k | 142.15k ± 838.43 | 29.10k ± 850.92 | 33.55k ± 427.98 | 2.23k ± 42.72 | 52.69k ± 448.91 |
| Tennis | 0.00 ± 0.00 | 20.77 ± 0.54 | 0.00 ± 0.00 | 0.00 ± 0.00 | -2.34 ± 0.58 | -1.69 ± 0.51 | 22.20 ± 0.17 |
| TimePilot | 34.33k ± 404.16 | 10.66k ± 159.67 | 55.38k ± 918.01 | 109.15k ± 4.05k | 33.82k ± 544.74 | 11.37k ± 222.24 | 42.26k ± 722.78 |
| Tutankham | 160.03 ± 1.36 | 213.78 ± 8.55 | 240.49 ± 9.66 | 187.71 ± 0.03 | 190.20 ± 1.81 | 120.27 ± 3.63 | 230.73 ± 0.84 |
| UpNDown | 76.44k ± 910.55 | 65.52k ± 9.48k | 422.67k ± 2.82k | 322.33k ± 3.00k | 240.21k ± 9.79k | 127.06k ± 2.61k | 264.48k ± 2.48k |
| Venture | 2.05k ± 12.84 | 1.50k ± 35.34 | 20.00 ± 0.00 | 1.99k ± 2.95 | 1.41k ± 24.61 | 1.50k ± 24.88 | 2.06k ± 3.76 |
| VideoPinball | 155.83k ± 3.47k | 341.15k ± 59.03k | 542.75k ± 15.87k | 392.46k ± 17.02k | 63.37k ± 4.76k | 117.67k ± 3.64k | 545.38k ± 76.91k |
| WizardOfWor | 31.94k ± 1.13k | 16.52k ± 1.16k | 19.41k ± 371.79 | 39.40k ± 406.34 | 9.68k ± 1.12k | 10.94k ± 529.67 | 58.41k ± 588.34 |
| YarsRevenge | 82.41k ± 1.41k | 70.73k ± 1.80k | 125.27k ± 1.07k | 139.64k ± 1.55k | 70.65k ± 600.98 | 75.95k ± 785.33 | 165.95k ± 245.07 |
| Zaxxon | 18.84k ± 311.75 | 12.58k ± 727.69 | 34.99k ± 233.45 | 45.80k ± 276.10 | 30.07k ± 262.07 | 13.28k ± 505.46 | 30.60k ± 310.47 |

*Table 5.* Full Performance of Atari Games.

