# OpenReview forum: "Reinforcement Learning with Discrete Diffusion Policies for Combinatorial Action Spaces"
_ICML.cc/2026/Conference — ICML 2026 regular_

### Official Review · Reviewer_dg3F · 2026-03-03

**Soundness:** 2
**Presentation:** 2
**Significance:** 2
**Originality:** 2
**Overall Recommendation:** 3
**Confidence:** 3

**Summary:**

This paper introduces a discrete diffusion approach on top of combinatorial action spaces. Combining with policy mirror descent trick, this approach has proven to be efficient on DNA sequence generation and atari game. At the same time, it also show some potential extending to multi agent setup.

**Compliance With Llm Reviewing Policy:**

Affirmed.

**Final Justification:**

I thank authors for the effort they made in addressing my concerns and questions. However, after reviewing the answers I still feel like this paper needs substantial improvement for acceptance.

**Key Questions For Authors:**

1, How is the combinatorial action space structure assumption used or aided for the policy optimization? Is it ok we only have a discrete action space assumption here?\
2, The problem setup in section 3.1 line 113-114 is bit confusing, is the problem originally defined in the multiagent setup?\
3, What's the difference between PMD and more general variational RL-based methods, such as AWR, MPO etc?\
4, For RL-D^2, is the training framework on-policy or off-policy? Authors mentioned that for the RKL training it needs to be on-policy, but for FKL there is off-policy adaption? \
5, For the IS ratio, is it possible to directly estimate from diffusion policy ELBO, which should be easy to get from the diffusion policy?

**Limitations:**

1, For the comparison on Atari game, there should be some comparisons with DreamerV3 or MuZero-like model-based SOTA algorithms. Current compared benchmarks are necessary but not sufficient. \
2, Besides, it would be better to include some naive diffusion policy attempt to motivate the necessity of discrete diffusion.\
3, Typo in line 19 Abstract section.\
4, Typo in line 70 in Related Work section.

**Strengths And Weaknesses:**

1, The contribution of the work seems unclear, since both the PMD and discrete diffusion structures are already there. And there are no further theoretical exploration on the combination of them.\
2, Paper writing and clarity in general need some improvement.

---

> ### Author Rebuttal · Authors · 2026-03-31
>
> ### W1: Contribution and Novelty
> While Policy Mirror Descent and discrete diffusion exist independently, our core contribution lies in theoretically and practically bridging them to solve high-dimensional combinatorial RL problems. We provide a unifying framework that derives tractable policy learning objectives under different divergence metrics (FKL and RKL) with respect to the optimal PMD target.
>
> Furthermore, we explore two distinct viewpoints for policy optimization within this framework:
>
> 1. Policy as a diffusion model: using the ELBO to bound the intractable likelihood, which reduces variance but introduces bias.
>
> 2. Policy as a backward process: formulating an augmented MDP composed of the original MDP and the diffusion steps, which provides an unbiased but higher-variance gradient.
>
> The implementation of these viewpoints into actionable, online RL algorithms (like FKL yielding an AWR-style objective, and RKL yielding an augmented-MDP PPO objective) is a highly non-trivial contribution to the diffusion-RL literature.
>
> ### Q1: Combinatorial Action Spaces Assumption
>
> The combinatorial structure is essential to our approach. Discrete diffusion exploits the factored structure directly: it generates actions by iteratively unmasking token positions, capturing cross-position dependencies (e.g., coordinated joint actions in multi-agent settings, temporally coherent macro-actions) that naive factorized policies with independency assumption cannot model. We validate this empirically against autoregressive baselines. A standard discrete action space (as in DQN or PPO) requires a flat categorical over all $K^N$ possible actions for $N$ dimensions with $K$ choices each. Standard RL simply cannot scale to this exponential combinatorial explosion.
>
> ### Q2: problem setup in section 3.1
>
> The multi-agent centralized controller is a special case of our Section 3.1 formulation: the state space includes the global state, and each component $i=1,...,K$ in the macro action space corresponds to the $i^{\rm th}$ agent's action. Our framework treats this joint action vector as the discrete sequence generated by diffusion.
>
> ### Q3: Differences with other variational RL methods.
>
> PMD is a general framework for regularized policy improvement; AWR and MPO are specific instantiations derivable from PMD under particular constraints (e.g., AWR is PMD with FKL). We use PMD to analytically derive the target policy, then show how matching this target via discrete diffusion connects back to practical algorithms.
>
> ### Q4: Training framework on-policy or off-policy
> Our implementations support both on- and off-policy.
>
> > Authors mentioned that for the RKL training it needs to be on-policy, but for FKL there is off-policy adaptation?
>
> Yes. RKL is on-policy only; FKL supports both. We use off-policy for Atari since its simulator is much slower than Google Football's.
>
> ### Q5: Estimating IS ratio by ELBO.
> We tested using the ELBO ratio to estimate the IS ratio on Google Football:
> | IS ratio estimator | pass & shoot | run pass & shot | 3 vs 1 | ct-easy | ct-hard | corner | 5 vs 5 | 11 vs 11 |
> | :--- | :--- | :--- | :--- | :--- | :--- | :--- | :--- | :--- |
> | **Augmented MDP** | **100 ± 0.0** | **98.5 ± 1.6** | **98.6 ± 1.0** | **99.1 ± 0.6** | **97.7 ± 2.2** | **96.3 ± 3.4** | **99.6 ± 0.4** | **99.8 ± 0.2** |
> | ELBO | 90.0 ± 10.0 | 62.6 ± 20.9 | 72.6 ± 14.0 | 7.9 ± 3.5 | 6.3 ± 0.6 | 13.4 ± 6.2 | 46.0 ± 13.4 | 0.0 ± 0.0 |
>
> The ELBO estimator underperforms because ELBO is a biased log-probability estimator, and this bias compounds when used for IS ratio estimation.
>
> ### L1: Comparisons with Model-Based SOTA
> We agree that DreamerV3 and MuZero are exceptional SOTA algorithms. The main scope of this paper is to study how policy parameterization affects performance, rather than how we leveraged data. Given the time budget we will add these baselines in the future revisions.
>
> ### L2: Naive Continuous Diffusion Baseline
> Thanks for pointing out. We will add this discussion. Continuous diffusion on discrete combinatorial spaces has three failure modes:
>
> 1. Vanishing gradients: mapping continuous outputs to discrete tokens (via argmax/rounding) kills gradient flow.
>
> 2. No action topology: continuous diffusion assumes ordinal structure (1.1 ≈ 1.2), but categorical actions lack a meaningful metric, so continuous noise is ill-defined.
>
> 3. Biased exploration: snapping Gaussian noise to a discrete grid warps probability mass and hurts exploration. Discrete diffusion avoids this by operating directly on transition matrices over categorical states.
>
> ---
>
> We appreciate your thoughtful review and hope our responses have clarified the concerns raised. We would be grateful if you would consider updating your score in light of these clarifications.

---

> > ### Author Rebuttal · Reviewer_dg3F · 2026-04-02
> >
> > Reviewer thank author's effort in addressing questions and concerns. However, reviewer is still concern about the novelty issue: 1, using ELBO to bound intractable ELBO seems to be the standard motivation for the variational RL, and diffusion denoising process naturally falls into this formulation. Besides, reviewer still feel like the comparison with current model-based approach to some extent is needed. Since the motivation of using diffusion policy is for its expressiveness and author also test trade-off between scalability and horizon, both resonates with core components of model-based approaches. This paper might be more completed with this integration to further support necessity of discrete diffusion policy under this context.

---

> > > ### Author Response · Authors · 2026-04-06
> > >
> > > We thank the reviewer for his feedback. We address the reviewer's two concerns as follows:
> > >
> > > **Variational RL:**
> > >
> > > We agree that using the ELBO to bound the likelihood is a standard technique in variational RL. However, the core novelty of our  framework is not the invention of the ELBO, but rather the non-trivial bridge of Policy Mirror Descent with masked discrete diffusion models.  From both a theoretical and practical viewpoint.
> > >
> > > Standard variational RL typically deals with simple, single-step distributions (like Gaussians or categorical distributions). In contrast, discrete diffusion involves a multi-step reverse generation process with a vast combinatorial space. Our contribution lies in successfully casting the PMD policy improvement step into a distributional matching problem for this specific, complex architecture.
> > >
> > > We also suggest more practical training techniques such as on policy diffusion steps sampling. Not only that, we also provide another optimization viewpoint by formulating an **augmented MDP** for the RKL objective, which optimizes the single-step backward process without computing the ELBO.
> > >
> > >
> > > These mechanisms are specific to the unique structure of diffusion models and represent a significant difference from standard variational methods.
> > >
> > > **The relationship with model based approaches:**
> > >
> > > We view model based RL and discrete diffusion policy as orthogonal and complementary, rather than mutually exclusive. MBRL focuses primarily on learning a world model to simulate data, plan, and improve sample efficiency. While our method focuses purely on policy parameterization. Our goal is to solve the bottleneck of representing and searching through massively high-dimensional, combinatorial action spaces where standard independent or autoregressive policies fail due to causal bias or exponential explosion.
> > >
> > > In fact, a discrete diffusion policy could theoretically be integrated into a MuZero or Dreamer style architecture as the underlying actor to handle combinatorial action spaces more effectively. To isolate and prove the expressiveness of our policy class, we benchmarked it against highly scalable,sota model free and multi agent methods (e.g R2D2, MAT, MAPPO) across domains like the 11 vs 11 Google Research Football scenario and Atari.
> > >
> > > Having said that, we agree that MBRL baselines will provide a more comprehensive picture. We will try to include empirical comparisons in the camera-ready revision.

---

### Official Review · Reviewer_v9ij · 2026-03-10

**Soundness:** 2
**Presentation:** 3
**Significance:** 3
**Originality:** 3
**Overall Recommendation:** 5
**Confidence:** 4

**Summary:**

The paper _Reinforcement Learning with Discrete Diffusion Policies for Combinatorial Action Spaces_ introduces a new framework for using discrete diffusion models as policies in reinforcement learning (RL). The paper frames policy learning as a distribution matching problem, training the discrete diffusion policy to match the improved, closed-form policy mirror descent (PMD) policy using distribution matching, either with a reverse KL divergence (RKL) or a forward KL divergence (FKL). The paper empirically studies the resulting algorithm, called RL-D², in a number of benchmark problems.

**Compliance With Llm Reviewing Policy:**

Affirmed.

**Final Justification:**

The rebuttal mostly addressed my concerns with the empirical study presented in the paper. My concern was that too few experimental repetitions were used (following the recommendation of Patterson et al (2024)), which would increase the rate of false positive conclusions. During the response period, the number of experimental repetitions throughout the paper were raised from 3/5 to 20 throughout the empirical study.

I believe that given these stronger empirical results, the paper provides novel, useful insights about discrete diffusion policies in RL and that the RL community would overall benefit from having these insights published.

**Key Questions For Authors:**

See my Main Argument above.

**Limitations:**

Yes

**Strengths And Weaknesses:**

# Main Argument

My concerns with this paper stem from the empirical study. Primarily, I am concerned that the claims made in the paper do not match the presented empirical evidence. Throughout the empirical study, the paper employs only three random seeds, calling into question the reported levels of statistical confidence _when they are reported_. For some of the experiments the paper does not present any estimates of statistical confidence, calling into questions the corresponding conclusions. Secondarily, a number of the experiments lack sufficient details -- or the experimental descriptions/figures are confusing -- which makes an accurate assessment of the paper difficult.

**Recommendation**: Reject

## Concerns About Claims Matching Evidence

My foremost concern with the paper is that the empirical study lacks a sufficient number of experimental repetitions for a rigorous statistical characterization of the algorithms presented, and that sometimes conclusions are drawn without presentation of statistical notions of confidence or variability.

**Too few random seeds are used** For a few experiments I was unable to find any indication of the number of random seeds used for the corresponding experiments. For the experiments on the MinAtar and Atari environment suites, the paper does mention that 3 random seeds were used, and so I will assume that 3 random seeds were used for all experiments. Likely, 3 seeds is too few for a robust statistical characterization of these algorithms [1, 2, 3], and while I appreciate that the kind of stratified sampling employed in a number of experiments increases this from 3 seeds to (3 seeds/env) * (4 envs) = 12 seeds, I am still concerned that this is too few experimental repetitions. It is well-established that deep reinforcement learning algorithms exhibit high variance and non-normal performance distributions [1, 2, 3]. Small sample sizes (even up to 10 seeds) can produce misleading estimates of central tendency and variability for such performance distributions, and can even lead to _statistically confidently incorrect conclusions_ for RL data as shown by Patterson et al. [1]. Further, the paper does not always provide indication of what kinds of confidence intervals are reported. Assuming these are standard errors or Student t confidence intervals, this raises further concerns as such estimates of statistical variation typically assume approximate normality, which is often not satisfied by return data in RL [1]. Because of this, I am suspicious of the reported levels of statistical confidence in the empirical study, as well as the associated claims made based on these.

While I recognize that the norm in deep RL is to use only a few experimental repetitions, this practice does not reflect scientific rigour, and the field is increasingly recognizing the need for higher empirical standards. I encourage the community, and this paper in particular, to work toward obtaining robust, statistically-accurate empirical results. This involves using a sufficient number of random seeds so that estimates of statistical confidence are accurate, as well as using estimates of confidence for which assumptions are satisfied. Doing this will strengthen this paper's interesting results, and I would be happy to re-evaluate my scoring of the paper in this case.


**Claims with overlapping confidence intervals** The paper makes claims about differences in performance when the reported confidence intervals overlap. I will go through a non-exhaustive list of examples here. First, in regards to Table 1, the paper states

> As shown in Table 1, RL-D² achieves substantially stronger performance in 4 out of 5 tasks in MinAtari.

But, we see that in fact RL-D² does not perform statistically significantly better than the baselines in 4 of the environments, since it is outperformed by DQN in Asterix and does not perform statistically significantly better than DQN in Freeway.

In regards to Table 2, the paper states:

> As presented in Table 2, our discrete diffusion policy variants achieve the highest mean win rates across all scenarios.

This claim is not supported by the presented empirical evidence, since the
presented confidence intervals of RL-D² overlap with the baselines for a number
of these scenarios. A more measured claim might be that "RL-D² often achieves a mean win rate at least as large as the baselines across all scenarios".

As another example, the paper states:

> Table 2 demonstrates that RKL typically outperforms FKL when trained with large sample sizes.

But again, this does not seem to be supported by the empirical evidence since
the confidence intervals for all of the Google Research Football benchmark
problems which were run for 1G steps overlap for RL-D² using both RKL and FKL (assuming this is what is meant by the term _large sample sizes_).


**Claims without any confidence intervals** Finally, the paper makes claims about differences in algorithm performance for results where no notion of statistical confidence is reported. For example, Figure 3, Figure 4 (right), Figure 5, Figure 8, etc. all do not report levels of statistical confidence, and yet the paper draws conclusions from these results. Validating such conclusions is impossible without any notion of statistical confidence.


## Other Concerns About Experiments

Finally, a number of the reported experimental results are concerning, or else
lacking sufficient experimental details to understand and accurately evaluate
them.

**MinAtar Results**. For the MinAtar results, how is it possible that RL-D² can achieve a reward of 178.9k in Space Invaders and 20.18K in Breakout? For Space Invaders, there are a maximum of 10 aliens at any given time. Assuming all aliens are shot on **each time step** with the default cutoff of 2,500 steps, the maximum reward should be 25,000 (+1 reward per alien). A similar argument hold for Breakout. Am I missing something here?

**Small Macro Action Lengths do well**. In Figure 4, the smallest macro action
length of 2 does very well. Doesn't this suggest that an even smaller
macro action length of 1 should be tried? Why does the small macro action
length do so well, and does this indicate that perhaps macro actions and these
kinds of diffusion policies are not needed?

Furthermore, in Figure 4 (left), it is shown that RL-D²
performs best with 2 or 4 macro action length, and that increasing the number to 8
or 16 decreases performance. Then in the same figure (right), it indicates that
increasing the macro action length (x-axis) increases performance. How is this
possible? Isn't this contradictory? Furthermore, the caption states that the
right subplot indicates that the proposed method does better with increasing
network size and data. It is not clear how this is shown in the figure at all
though, since as far as I can tell the network size and data is not ablated
here.

**Confusion about Figure 5** The caption to Figure 5 and main text indicate
that Figure 5 shows a comparison in mean episode return of RL-D² with length 16 and 8 macro actions as a function of model parameters, data samples, and training time. But, it is not clear in the figure which line refers to the length 8 macro action, since in the main text it states that the baseline curve is IMAPLA-Macro. This makes it

# Small Things
* In Figure 6, the caption and main text reference _remasking_, but the legend does not clearly indicate which of the plotted lines this refers to. It would be good to update the legend to make it a bit more clear.
* Paragraph starting on L260, col 2 references Figure 2 (left) and (right), but this should be (top) and (bottom).
* Figure 2 caption references left/right subfigures, but this should be top/bottom
* The x-axis tick labels in Figure 2 (top) specify RL-D²−32 and RL-D²-200, but as far as I can tell, these names have not been defined
* Line 250: "within a fix mount of time" → "within a fixed amount of time"
* Line 69: the citation "()zekri2025fine" needs to be fixed
* Line 19 (abstract): "...effective policy improvement and   By leveraging..." → "...effective policy improvement. By leveraging..."

# References

[1] Andrew Patterson, Samuel Neumann, Martha White, Adam White. _Empirical Design in Reinforcement Learning_. JMLR, 2024.

[2]  Peter Henderson, Riashat Islam, Philip Bachman, Joelle Pineau, Doina Precup, David Meger. _Deep Reinforcement Learning that Matters_. AAAI, 2018.

[3] Cédric Colas, Olivier Sigaud, Pierre-Yves Oudeyer. _How Many Random Seeds? Statistical Power Analysis in Deep Reinforcement Learning Experiments_.

---

> ### Author Rebuttal · Authors · 2026-03-31
>
> We appreciate the reviewer’s detailed and rigorous comments and the effort to push the whole RL community towards rigorous evaluation of the RL algorithm. Please find our reponses below and follow the links for a collection of the updated figures.
>
> ## Q1: Statistical rigor concern
>
> ### Q1.1: Too few random seeds are used
>
> For the Atari tasks, the performance comparison experiment runs 3 seeds with 56 tasks, for a total of 168 runs. For a more concrete evaluation, we
>
> 1. re-evaluate the performance using different aggregation methods, mean, median and IQM, with confidence intervals (CIs) by stratified bootstrap sampling shown in [*Figure 1, 2 and Table 1*](https://anonymous.4open.science/r/icml26rebfig-65BE/figures.pdf).
>
> 2. [*Figure 3*](https://anonymous.4open.science/r/icml26rebfig-65BE/figures.pdf) evaluates the statistics of performance differences according to [1], while the lower bounds of the stratified sampling CIs are all above 0.
>
> We clarify that we use 5 random seeds for the Google Football tasks (explained in Figure 11, and we will state it clearly in the main text in our revision.)
>
> Additionally, we will run additional seeds for the final revision of the paper, as the reviewer suggests.
>
> ### Q1.2: Claims with overlapping confidence intervals
>
> We appreciate the reviewer's suggestion for precise language.
>
> - **MinAtar**
>
>   We will modify the MinAtar claim to *As shown in Table 1, RL-D² achieves substantially stronger performance in 2 out of 5 tasks in MinAtar*.
>
> - **Google Football**
>   > our ... achieve the highest mean win rates across all scenarios.
>
>   We note that our claim refers to the observed mean, which is factually the highest in all scenarios (except 11 vs 11 sm where only FKL succeeds). Confidence intervals overlap with a baseline in only 2 of 9 scenarios (run pass \& shot and corner with MAT), while in the majority, especially in the most challenging settings (ct-hard, 5 vs 5, 11 vs 11), the gap is well beyond confidence intervals. We believe the overall evidence strongly supports the effectiveness of our method.
>
>   > confidence intervals for all benchmarks ... overlap for RL-D² using both RKL and FKL.
>
>   we would like to clarify that all the tasks except 11 vs 11 SM are considered as *large sample size* as the performance levels. RKL and FKL's confidence intervals overlapped in 4 tasks while RKL outperforms FKL in the other 4 more challenging tasks. we will change our claim to *FKL and RKL gets comparable performance, while RKL typically outperforms FKL in those challenging tasks.*
>
> ### Q1.3: Claims without any confidence intervals
>
> We re-plotted original Figure 3, Figure 4 Right, and Figure 5 using 95\% CIs from stratified bootstrap sampling from runs with multiple tasks in [*Figure 1, 4, 5*](https://anonymous.4open.science/r/icml26rebfig-65BE/figures.pdf). We will run more random seeds and update all the following plots in the appendix with proper CIs.
>
> ## Q2 Other Concerns
>
> - **MinAtar performance.**
>
>   According to the documentation, for Breakout, the termination happens when the ball hit bottom of the screen. For Space Invaders, the termination happens when an alien or a bullet hits the player. No strict 2500-step limits are set for these tasks. Therefore, the high score indicates RL-D$^2$ can learn long-horizon planning with macro-actions.
>
> - **Small Macro Action Lengths do well**
>
>   We would like to clarify that macro-actions were selected primarily as a canonical example of a combinatorial action space with adaptive difficulties. Our primary goal was to demonstrate the scalability of our method in these challenging combinatorial discrete spaces, rather than to argue that long-horizon open-loop execution is the better strategy for Atari specifically.
>
>  Additionally, we proposed the planner-based variant of RL-D$^2$
>  (Appendix E.3 with replotted results in [*Figure 6*](https://anonymous.4open.science/r/icml26rebfig-65BE/figures.pdf)), which tackles the risk of compounding errors, where the policy functions work as a receding horizon planner. In this setup, the model generates a multi-step macro-action but executes only the first action before replanning.
>
>
> ## Other questions
> - **Confusion about Figure 4.** We listed the detailed setup in Appendix D.3, and we will add links in the revision.
>
> - **Confusion about Figure 5.**
> Thanks for pointing that out. The *baseline IMPALA-macro* in the main text means baselines in Figure 4. We fix in our replotted [*Figure 5*](https://anonymous.4open.science/r/icml26rebfig-65BE/figures.pdf).
>
> - **Legend of Figure 6.**
>   We update the plot in [*Figure 7*](https://anonymous.4open.science/r/icml26rebfig-65BE/figures.pdf).
>
>
> [1] Deep reinforcement learning at the edge of the statistical precipice.
>
>  ---
>
> We appreciate your thoughtful review and hope our responses have clarified the concerns raised. We would be grateful if you would consider updating your score in light of these clarifications.

---

> > ### Author Rebuttal · Reviewer_v9ij · 2026-04-03
> >
> > Thank you for your detailed response and clarifying remarks.
> >
> > I would like to clarify that my remarks in my original review on using too few random seeds were not specifically for the Atari results across the entire suite (as you rightly pointed out, many experimental repetitions exist here) but rather for the per-environment MinAtar results (Table 1) and the Atari subset figures (Figure 4-6). Although the percentile bootstrap confidence interval does not make any distributional assumptions, it does assume that sampling from the dataset well-approximates sampling from the underlying distribution (hence, it is not a solution to using too few experimental repetitions). Patterson et al. [1] showed that with even up to **10 experimental repetitions**, many different methods of estimating statistical confidence, including the percentile bootstrap, can produce _statistically confidently incorrect_ conclusions. Because of this, I still think the corresponding experiments need more repetitions (e.g. > 20) in order to be statistically rigorous.
> >
> >
> > > As presented in Table 2, our discrete diffusion policy variants achieve the highest mean win rates across all scenarios.
> >
> > Although the observed sample mean is the highest in all scenarios, I do not think that this observation itself is of much use. The main point of statistically characterizing these algorithms is to infer statistics of the underlying distribution which is why we use confidence intervals in the first place. Statements which discriminate between algorithm performances imply this kind of statistical characterization, which is what my points referred to in my original review. Nevertheless, while I agree that the confidence intervals presented do not overlap with the baselines in a number of scenarios, I am concerned that the reported levels of confidence are not accurate for this problem suite due to the use of too few random seeds (see my comments above). Again, Patterson et al. [1] demonstrated that when using few random seeds, it is not unlikely to end up with a number of "lucky" seeds which produce overly tight confidence bounds.
> >
> > In regards to the replotted Figure 4, it does appear that the confidence intervals for RL-D2 and IMPALA overlap significantly for different action lengths, which does call into question the associated claims in the paper submission. Given my previous comments with regards to confidence intervals, I do think that 12 experimental repetitions are likely too few for this setting.

---

> > > ### Author Response · Authors · 2026-04-06
> > >
> > > We thank the reviewer for the clarification. In response, we increased the number of random seeds for MinAtar, Google Research Football, and Atari to **20** and recomputed the bootstrap confidence intervals. These additional runs preserve the same qualitative trends as in our rebuttal, but we agree that the claims should be stated more carefully. In the following tables, each entry reports the sample mean over 20 random seeds, with the 95% bootstrap confidence interval shown in brackets.
> > >
> > > **MinAtar**
> > >
> > > | | ASTERIX | BREAKOUT | FREEWAY | SEAQUEST | SPACE INVADERS |
> > > | :--- | :--- | :--- | :--- | :--- | :--- |
> > > | DQN | **284.38 [112.88, 455.88]** | 180.36 [64.23, 296.49] | **60.44 [58.46, 62.42]** | **109.20 [81.23, 137.17]** | 1.16k [0.51k, 1.81k] |
> > > | DQN-MACRO | 43.05 [32.68, 53.42] | 315.85 [111.90, 519.80] | **58.42 [53.51, 63.33]** | **166.65 [109.25, 224.05]** | 769.86 [258.27, 1281.45] |
> > > | IMPALA | 24.78 [17.38, 32.18] | 1.02 [1.02, 1.02] | **41.01 [26.90, 55.12]** | 44.55 [30.95, 58.15] | 47.71 [47.71, 47.71] |
> > > | IMPALA-MACRO | 21.29 [12.68, 29.90] | 7.21 [6.78, 7.64] | **54.09 [44.37, 63.81]** | 55.42 [32.49, 78.35] | 33.18 [33.18, 33.18] |
> > > | RL-D² (ours) | 52.38 [44.59, 60.17] | **20.58k [13.47k, 27.46k]** | **59.98 [57.70, 62.26]** | **164.20 [109.21, 219.19]** | **184.3k [62.56k, 305.9k]** |
> > >
> > > **Google Football**
> > >
> > > | Method | PASS & SHOOT | RUN PASS & SHOOT | 3 vs 1 | CT-EASY | CT-HARD | CORNER | 5 vs 5 | 11 vs 11 | 11 vs 11 SM |
> > > | :--- | :--- | :--- | :--- | :--- | :--- | :--- | :--- | :--- | :--- |
> > > | AUTOREGRESSIVE | 79.4 [44.2, 94.5] | 68.9 [36.4, 89.3] | 74.1 [43.4, 91.7] | 70.8 [43.0, 89.5] | 35.2 [0.5, 66.5] | 0.0 [0.0, 0.0] | 72.5 [46.0, 90.1] | 33.1 [19.8, 46.4] | 0.0 [0.0, 0.0] |
> > > | RL-D² RKL (ours) | **100.0 [100.0, 100.0]** | **97.9 [96.3, 99.1]** | **99.2 [98.5, 99.7]** | **98.5 [97.7, 99.2]** | **98.2 [96.3, 99.4]** | **97.1 [96.3, 98.9]** | **99.1 [98.7, 99.5]** | **99.4 [99.0, 99.7]** | 0.0 [0.0, 0.0] |
> > > | RL-D² FKL (ours) | **98.4 [97.3, 99.3]** | **99.2 [98.7, 99.6]** | 96.8 [94.8, 98.5] | **97.4 [96.2, 98.6]** | **96.5 [93.7, 98.6]** | 94.1 [91.7, 96.1] | 97.9 [96.8, 98.5] | 98.1 [97.1, 99.0] | **68.5 [62.0, 75.0]** |
> > > | MAT | **96.8 [94.6, 98.6]** | **97.6 [96.3, 98.8]** | 91.8 [90.6, 93.0] | 88.5 [86.4, 90.6] | 87.1 [83.5, 90.7] | 94.6 [92.1, 95.9] | 92.9 [91.9, 93.9] | 91.3 [88.9, 93.7] | 10.2 [7.4, 13.0] |
> > > | MAPPO | 98.8 [98.4, 99.2] | 74.5 [71.6, 77.4] | 92.6 [91.0, 94.2] | 71.4 [68.3, 74.5] | 64.2 [62.1, 66.3] | 54.7 [50.2, 59.2] | 94.8 [93.1, 96.5] | 53.9 [50.9, 56.9] | 4.8 [4.5, 5.1] |
> > >
> > > These results support the same revised conclusions as in our rebuttal:
> > >
> > > - In MinAtar, RL-D² shows clearly stronger performance on 2 of 5 tasks.
> > > - In Google Football, FKL and RKL are overall comparable; RKL is typically stronger on more challenging tasks, while FKL performs better on 11 vs 11 SM despite using only 1/10 as many samples as standard 11 vs 11.
> > >
> > > **Atari**
> > >
> > > - We replotted Figures 4 and 5 with 8 additional runs (20 runs in total) [here](https://anonymous.4open.science/api/repo/icml26rebfig-65BE/file/figures_v2.pdf?v=29167345).
> > >
> > > We agree that in Figure 4, the confidence intervals overlap with the baselines for macro action lengths 2, 4, and 8. We will therefore revise the claim to:
> > >
> > > > *RL-D² performs at least as well as IMPALA-Macro up to macro action length 8, and outperforms IMPALA-Macro at macro action length 16.*
> > >
> > > For the remaining Atari results, including Figure 6 and Table 1, which serve as supporting evidence, we will expand this evaluation in the revision and revise the conclusions accordingly.
> > >
> > > ---
> > >
> > > Overall, we will revise the paper using these narrower claims to avoid statistical overstatement. We hope these additional experiments and revisions could further address the reviewer’s concerns and support a more positive overall assessment.

---

### Official Review · Reviewer_qBUD · 2026-03-12

**Soundness:** 3
**Presentation:** 3
**Significance:** 3
**Originality:** 3
**Overall Recommendation:** 4
**Confidence:** 3

**Summary:**

In this paper, the authors propose RL-D^2, a novel online RL framework that employs discrete diffusion models as expressive policies for large combinatorial action spaces. By framing policy improvement as distributional matching to a stable target derived from policy mirror descent (PMD), the method decouples RL optimization from representation learning, yielding robust training dynamics.
The authors propose forward and reverse KL objectives, with practical approximations for the intractable likelihood ratio in the reverse KL case, and introduce on-policy diffusion learning to align training with the policy’s generative process.
Extensive experiments across DNA sequence optimization, macro-action RL in MinAtar/Atari, and multi-agent football demonstrate state-of-the-art performance and superior efficiency compared to baselines.

**Compliance With Llm Reviewing Policy:**

Affirmed.

**Final Justification:**

Most of my concerns have been addressed.  The proposed method may be good at addressing certain kinds of combination problems, particularly those that can be translated as RL problems with discrete actions.  The experiments show that it can scale up to  high dimensional problems.   Considering this,   I lean towards accepting the paper and keeping my original score.

**Key Questions For Authors:**

(1) Could the authors provide a discussion or comparison of the proposed method with discrete black-box solvers?

(2) Could the authors discuss more about which family of problems the proposed method can address well and which family it cannot.   What is the boundary and feasible region of the proposed method?

**Limitations:**

yes

**Strengths And Weaknesses:**

Strengths

(1) Soundness:  The paper is technically sound.

(2) Presentation:  The paper is well written and well structured.

(3) Significance:  The combinatorial optimization is an important problem with many real-world applications.  The empirical results  on  extensive  benchmark problems are strong.

(4) Originality:  The proposed RL-D^2  framework that employs discrete diffusion models as policies for reinforcement learning in large combinatorial action spaces is novel.


Weaknesses

(1) Sample/Query Efficiency of Diffusion Training:
In many real-world applications of combination optimization, e.g., chip design and drug discovery, the query is expensive.  As the diffusion model involves large parameters, the training may lead to a high sample/Query cost.  I am not sure about the query efficiency of the proposed method compared with discrete black-box solvers.

(2) Combinatorial optimization problems are challenging, and many of them are NP.  I am not sure which family of problems the proposed method can address well and which family it cannot.

---

> ### Author Rebuttal · Authors · 2026-03-31
>
> We thank the reviewer for the positive comments. Below are our responses.
>
> ### W1/Q1: Sample/Query Efficiency of Diffusion Training
>
> > *In many real-world applications of combination optimization, e.g., chip design and drug discovery, the query is expensive. As the diffusion model involves large parameters, the training may lead to a high sample/Query cost. I am not sure about the query efficiency of the proposed method compared with discrete black-box solvers.*
>
> > *Could the authors provide a discussion or comparison of the proposed method with discrete black-box solvers?*
>
> **A1:**
> At a high level, RL interacting with an environment is itself a form of black-box optimization, where the transition dynamics and reward function act as the black box. That said, our RL-D$^2$ framework has several practical advantages over standard discrete black-box solvers (e.g., evolutionary algorithms, zeroth-order methods):
>
> 1. **Multi-step dynamics and structured exploration.** Traditional black box solvers typically optimize for single-step, static objectives. In contrast, RL is explicitly designed to leverage the transition dynamics of multi step, sequential environments like Atari and Google Research Football. In large combinatorial action spaces, standard solvers struggle to explore efficiently. Our discrete diffusion policies adapt effectively to these large spaces, providing a highly expressive prior that drives structured and targeted exploration.
>
> 2. **Generalization to new states.** Most discrete black-box solvers optimize from scratch for each problem instance (e.g., finding one good configuration). Our method instead learns a state-conditioned policy, so the diffusion model can transfer what it has learned and produce good combinatorial actions for new, unseen states without re-solving from scratch.
>
> 3. **Scaling to high dimensions.** Evolutionary and other heuristic methods tend to scale poorly to high-dimension problems due to the curse of dimensionality. By framing the optimization as distributional matching via PMD and leveraging the representation power of deep neural networks, our method scales highly effectively. We demonstrate this empirically through our strong performance and scaling results across high-dimensional benchmarks.
>
> 4. **Off-policy learning.** Our FKL variant supports off-policy training, so we can reuse data from earlier runs or existing datasets, which further improves query efficiency.
>
> We will add a discussion of this comparison in the paper.
>
>
> ### W2/Q2: Boundary and feasible region of the proposed method
>
> > *Combinatorial optimization problems are challenging, and many of them are NP. I am not sure which family of problems the proposed method can address well and which family it cannot.*
>
> > *Could the authors discuss more about which family of problems the proposed method can address well and which family it cannot. What is the boundary and feasible region of the proposed method?*
>
> **A2:** Our method works best when:
>
> 1. the combinatorial action space can be written as a fixed-length sequence of discrete tokens (e.g., multi-dimensional discrete actions, macro-actions, joint agent actions),
> 2. a pretrained generative model is available or can be trained from data to provide a meaningful prior over the space,
> 3. the goal is to find a distribution of diverse, high-quality solutions rather than a single optimum.
>
> Our three experimental domains fit this well: DNA sequence design (pretrained on genomic data), macro-action RL (structured temporal action sequences), and multi-agent coordination (joint action spaces). On the other hand, our method is less suited for classical NP-hard combinatorial optimization problems (e.g., TSP, bin packing) where hard feasibility constraints must be met exactly, and where specialized solvers built over decades already take advantage of problem-specific structure. We see our work as complementary to these approaches, targeting the broad class of RL and generative design problems where expressive, distributional policies over combinatorial spaces are needed.
>
> ---
>
> We appreciate your thoughtful review and hope our responses have clarified the concerns raised. We would be grateful if you would consider updating your score in light of these clarifications.

---

> > ### Author Rebuttal · Reviewer_qBUD · 2026-04-03
> >
> > Thanks for the authors' reply. Most of my concerns have been addressed.

---

### Official Review · Reviewer_ymi9 · 2026-03-13

**Soundness:** 3
**Presentation:** 2
**Significance:** 3
**Originality:** 2
**Overall Recommendation:** 3
**Confidence:** 2

**Summary:**

This paper proposed a discrete diffusion policy training framework for RL problem with large, combinatorial action spaces.

**Compliance With Llm Reviewing Policy:**

Affirmed.

**Key Questions For Authors:**

- The objective function used in the paper seems different from the score-matching/flow-matching style. Does the discrete manner break this?
- Will the discrete setting in the problem bring much difficulty in applying the conventional continuous diffusion policy?

**Limitations:**

The main novelty appears to lie more in adapting existing diffusion-based policy fitting ideas to a new setting than in introducing a fundamentally new policy optimization principle.

**Strengths And Weaknesses:**

Strengths:
- The focused problem is important, which is hard to solve for classical RL methods.
- The presentation is clear, and the discussion of the trade-off between FKL and RKL is useful.


Weaknesses:
- The paper seems being prepared in a short time, there is some typos, like  ``()zekri2025fine" around line 070, and a bad equation format in Eq. (4).
- The novelty seems limited from method perspective. The core idea of the proposed method is already well-established in diffusion-RL track. For example, the optimal policy is given by an analytical distribution [1], then diffusion policy can be applied to fit this policy [2,3], offline and online are both valid. From this sense, the novelty seems come from the application in a different domain.

[1].Peng X B, Kumar A, Zhang G, et al. Advantage-weighted regression: Simple and scalable off-policy reinforcement learning[J]. arXiv preprint arXiv:1910.00177, 2019.

[2]. Hansen-Estruch P, Kostrikov I, Janner M, et al. Idql: Implicit q-learning as an actor-critic method with diffusion policies[J]. arXiv preprint arXiv:2304.10573, 2023.

[3]. Frans K, Park S, Abbeel P, et al. Diffusion guidance is a controllable policy improvement operator[J]. arXiv preprint arXiv:2505.23458, 2025.

---

> ### Author Rebuttal · Authors · 2026-03-31
>
> We appreciate the reviewer’s comments. Please find our replies to your comments below.
>
> ### Q1: Limited novelty from method perspective
>
> > *The novelty seems limited from method perspective. The core idea of the proposed method is already well-established in diffusion-RL track. For example, the optimal policy is given by an analytical distribution [1], then diffusion policy can be applied to fit this policy [2,3], offline and online are both valid. From this sense, the novelty seems come from the application in a different domain.*
>
> We would like to clarify that our contribution is a general framework unifying different policy learning objectives under different distances w.r.t. the target optimal mirror descent policy in Eq. (3), and evaluating their empirical trade-off. In addition, for each distance, we may choose between two viewpoints of policy optimization:
>
> 1. **Policy as a Diffusion Model:** The likelihood is intractable, and we use the Evidence Lower Bound (ELBO) to bound it. This approach reduces variance but adds bias.
>
> 2. **Policy as the Backward Process:** This approach solves an augmented MDP composed of the original MDP and the diffusion states. This is an unbiased approach but is significantly noisier.
>
> Each combination of these viewpoints with different distances produces a distinct online RL algorithm. In our paper, we investigated two specific mixtures: **FKL with policy as a diffusion model** (producing an online objective aligned with AWR in the offline setting) and **RKL with policy as a backward process** (yielding a PPO-like objective w.r.t. the backward process). This modularity is a key contribution to the diffusion-RL literature.
>
> The focus of our paper is different from the papers referred to by the reviewer:
>
> 1. The papers all focus on offline RL. We focus on online RL, focusing on specific trade-offs such as exploration vs. exploitation, data efficiency vs. stability, etc. Directly using offline RL algorithms for online RL might fail due to not considering these trade-offs.
>
> 2. The references all focused on continuous action space. Certain adaptations and challenges in switching from continuous action space to discrete space are discussed in Q2, and the difference between continuous and discrete diffusion is discussed in Q3.
>
> ### Q2. Objective function v.s. score/flow matching
>
> **A2:** We thank the reviewer for this question and the opportunity to clarify. The discrete setting does not break the score-matching or flow-matching. Rather, it adapts it to discrete domain.
>
> The reviewer is likely referring to continuous flow-matching or diffusion models, where the objective typically takes the form of a MSE between the noise (or clean sample) and its predictor. However, in discrete diffusion or discrete flow-matching [1], the loss naturally takes the form of cross-entropy between the clean sample and its predictor.
>
> In our RL objective, our goal is to estimate and maximize the log-likelihood of the policy (represented by the diffusion model). To do this, we optimize the ELBO, which bounds the log-likelihood. The ELBO is composed of the sum of KL divergences between the true posterior and the estimated posterior. In the context of masked discrete diffusion, these KL divergences mathematically reduce to a cross-entropy objective. Therefore, our objective function remains fundamentally aligned with standard discrete flow-matching.
>
> [1] Discrete Flow Matching, Gat et al. NeurIPS 2024.
>
> ### Q3: Applying conventional continuous diffusion policy to discrete space
>
> > *Will the discrete setting in the problem bring much difficulty in applying the conventional continuous diffusion policy?*
>
> Applying continuous diffusion policy in the discrete setup will have multiple failure modes:
>
> - **Vanishing gradients.** Mapping from continuous space to discrete space will prevent the agent from seeing how changing its internal parameters would change the resulting discrete action, causing vanishing gradients.
>
> - **Lack of meaningful action topology.** Continuous action spaces assume an inherent ordinal relationship (e.g., 1.1 is closer to 1.2 than to 5.0). Discrete spaces often do not have this property, which significantly hurts interpolation and generalization.
>
> - **Biased exploration.** Continuous algorithms typically explore by adding Gaussian noise. When this noise is mapped onto a discrete grid, the probability mass becomes highly distorted and causes exploration failure.
>
> All aforementioned difficulties get worse when the space becomes combinatorial, motivating a discrete diffusion policy for a combinatorial discrete action space.
>
> We thank the reviewer for pointing out the typos in our paper. We will fix them in the final revision.
>
> ---
>
> We appreciate your thoughtful review and hope our responses have clarified the concerns raised. We would be grateful if you would consider updating your score in light of these clarifications.

---

### Decision · Program_Chairs · 2026-04-30

**Decision:**

Accept (regular)

**Comment:**

This paper proposes a novel RL strategy for training diffusion policies with large combinatorial action spaces. The reviewers generally agreed that the paper is tackling an interesting and important problem. There are some lingering concerns about novelty, but the authors provided an argument as to how exactly their approach differs from previous ones. There were also some concerns about the statistical methodology in the experiments, but these concerns were addressed during the rebuttal period.